# CausalKANs: interpretable treatment effect estimation with Kolmogorov-Arnold networks

## Abstract

Deep neural networks achieve state-of-the-art performance in estimating heterogeneous treatment effects, but their opacity limits trust and adoption in sensitive domains such as medicine, economics and public policy. Building on well-established and high-performing causal neural architectures, we propose *causalKANs*, a framework that transforms neural estimators of *conditional average treatment effects* (CATEs) into Kolmogorov–Arnold Networks (KANs). By incorporating pruning and symbolic simplification, causalKANs yields interpretable closed-form formulas while preserving predictive accuracy. Experiments on benchmark datasets demonstrate that causalKANs perform on par with neural baselines in CATE error metrics, and that even simple KAN variants achieve competitive performance, offering a favorable accuracy–interpretability trade-off. By combining reliability with analytic accessibility, causalKANs provide auditable estimators supported by closed-form expressions and interpretable plots, enabling trustworthy individualized decision-making in high-stakes settings. We release the code for reproducibility.

## 1 Introduction

Estimating individual treatment effects from observational tabular data underpins high-stakes decisions in personalized medicine (Kent et al., 2018; Sanchez et al., 2022), public policy (Imai and Strauss, 2011), and economics (Manski, 2004), where interventions must be tailored beyond population averages (Wager and Athey, 2018). As personalized decision-making becomes the norm, accurately recovering conditional average treatment effects (CATEs) is indispensable for policy targeting and individualized care (Curth et al., 2024). However, accuracy alone is insufficient: regulatory frameworks (GDPR Art. 22; EU AI

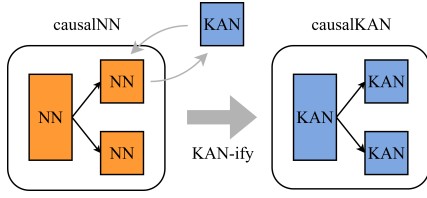

Figure 1: KAN-ification.

Act transparency for high-risk AI) and clinical practice increasingly discourage opaque models in consequential settings (European Parliament and Council, 2016; 2024; Goodman and Flaxman, 2017). Indeed, limited interpretability remains a barrier to the clinical adoption of machine learning (ML) systems (Amann et al., 2020; Tonekaboni et al., 2019). Yet contemporary state-of-the-art CATE estimators often rely on deep neural networks (Shalit et al., 2017; Shi et al., 2019), which we call causalNN, achieving strong performance but hindering auditing and trustable deployment.

We address this gap proposing *causalKANs*: a practical framework that transforms, or *KAN-ifies* (see Fig. 1), neural tabular CATE estimators into *closed-form*, auditable models by replacing their subnetworks with Kolmogorov–Arnold Networks (KANs) (Liu et al., 2024a;b). KANs parameterize one-dimensional edge functions (splines) and compose them via sums (and, in variants, products), enabling post-training simplification without departing from the trained predictor. Our interpretability notion is operational: after training, we apply edge-activity regularization, validation-guided pruning, and auto-symbolic substitution of learned splines with simple atoms to produce executable expressions for the potential outcomes and their difference (the CATE).

Concretely, our **contributions** are threefold. **First**, we introduce a **model-agnostic framework** for constructing KAN-based potential-outcome models from established causal neural architectures (e.g., metalearners (Künzel et al., 2019), TARNet (Shalit et al., 2017), DragonNet (Shi et al., 2019)) while

reusing their training objectives (see Fig. 1 for a sketch). **Second**, we propose a complete pipeline to achieve interpretability that includes pruning, symbolic substitution, and representation tools. **Third**, we present a comprehensive **empirical study** that **i)** compares performance of causalKANs on several well known benchmark datasets (IHDP (Hill, 2011), ACIC (Dorie et al., 2019), NSLM (Carvalho et al., 2019), NEWS (Johansson et al., 2016) and TCGA (Schwab et al., 2020)), and **ii)** assess the identification of causal equations with known synthetic datasets. The objective of benchmarking is to demonstrate that the use of KANs do not decrease the performance in causal metrics, namely, causalKANs are competitive with respect to causalKAN , *not necessarily better*. For example, in Fig. 2 we can observe that we have not found statistical difference between T-KAN and DragonNet, which are the best models for IHDP A. In addition, the aim of **ii)** is to give practical reasons to think that if the true causal equations lie in the space that causalKANs can model, then the causal equation can be recovered up to an algebraic equivalence.

Our study positions causalKANs as domain-agnostic, accuracy-preserving, and inspection-ready CATE estimators: they retain the flexibility of deep architectures yet yield executable formulas that make effect modifiers and interactions explicit, aligning with emerging requirements for trustworthy, human-auditable decision support.

Empirically, causalKANs provide a favorable accuracy–interpretability frontier: simple heads (often additive or one hidden KAN layer) achieve *competitive* metrics across benchmarks, while additional depth rarely improves accuracy and consistently reduces formula compactness.

We present the preliminaries in §3, the causalKANs pipeline and instantiations in §4, and reports ablations, comparisons, and interpretability results in §5.

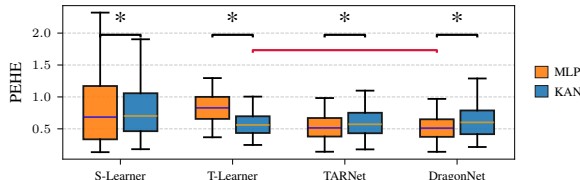

Figure 2: Overall, causalKANs achieves similar PEHE (lower is better) than causalNNs in IHDP A. * means statistical difference $p < 0.05$ in a Wilcoxon paired test. Red brace compares the best causalKAN  with the best causalNN, without indicating statistical difference.

## 2  RELATED WORK

Kolmogorov-Arnold netwoks, as interpretable and parameter efficient alternatives of multi-layer perceptrons (MLP), have experienced a great growth in the last year, with the adaptation of existing technologies as convolutional networks (Bodner et al., 2024), residual networks (Yu et al., 2024), quantum networks (Wakaura et al., 2025), or autoencoders (Moradi et al., 2024). In the same manner, they have had an impressive adoption in domains where transparency is critical, as predicting risks in cardiovascular diseases (Al Bataineh et al., 2025) in healthcare (Almodóvar et al., 2025; Pendyala and Venkatachalam, 2025); predicting volatility in finances (Cho et al., 2025); predicting biomarkers (Alharbi et al., 2025), or predicting physics in power systems (Shuai and Li, 2025).

Interpretability is widely regarded as essential for the adoption of ML in sensitive domains (Doshi-Velez and Kim, 2017). Methods for improving interpretability based on KANs have been used in critical domains, such as Kolmogorov-Arnold Additive Models (KAAMs) (Almodóvar et al., 2025), which are shallow KANs that yields nonlinear additive models, which are well known as interpretable flexible estimators, as Neural Additive Models (NAMs) (Agarwal et al., 2021) or generalized additive models (Caruana et al., 2015). A closely related line is *symbolic regression* (SR), which searches over expression trees to recover closed-form models, classically via genetic programming (Schmidt and Lipson, 2009) and can produce highly concise formulas; however, SR is also computationally intensive and scales poorly (Zhang et al., 2022), while KANs, optimized by gradient descent, have been reported to be parameter-efficient and fast to optimize in practice (Liu et al., 2024b).

The need for interpretability is equally pressing in causal inference, where estimators are expected not only to predict counterfactuals but also to justify treatment decisions (Athey and Imbens, 2016). Early approaches such as causal trees (Athey and Imbens, 2016) provided transparent rule sets, and linear regressors were interpretable through their coefficients as causal parameters (Hahn et al., 2018). However, partially interpretable algorithms like causal forests (Foster et al., 2011) and BART (Hill, 2011) offered better predictive accuracy, while neural networks advanced performance even further (Johansson et al., 2016; Schwab et al., 2018; Shalit et al., 2017; Yao et al., 2018; Yoon et al., 2018) at

the cost of opacity. As a result, interpretability has often been relegated to a secondary concern in causal estimation (Rudin, 2018).

Bridging the trade-off between accuracy and transparency remains challenging. Some promising attempts include attention-based transformers highlighting confounder importance (Zhang et al., 2023), causal rule learners that yield decision rules but rely on black-box induction (Bargagli-Stoffi et al., 2020; Wu et al., 2023), fused lasso regression-based methods producing interpretable effect curves (Padilla et al., 2021), and, importantly, NAM-based estimators of average treatment effects (Chen et al., 2025). *Model distillation* approaches, such as training a surrogate on top of the work of Shalit et al. (2017) (Kim and Bastani, 2019), offer another path but depend on surrogate fidelity. Yet, adressing causal inference and interpretability together requires futher work (Moraffah et al., 2020). In this context, KAN-inspired models such as our approach are promising because they combine the interpretability of additive structures with an expressive power undistinguishable of that of neural networks.

To the best of our knowledge, we are the second work that employs KANs for causal estimation, after KANITE (Mehendale et al., 2025), which replaces MLP backbones with KANs to estimate individual treatment effects under multiple treatments, using integral probability metrics/entropy-balancing variants and reporting gains in causal inference metrics over strong baselines. Although this paper already performs a KAN-ification, it does not provide a systematic method for carrying it out, and its emphasis is predictive accuracy; it neither targets interpretability nor provides closed-form effect functions or visually interpretable plots. Instead, our work focuses on extracting human-readable causal effect formulas.

Last, we also want to compare our proposal with the NAM-based approach of Chen et al. (2025). First, while NAMs offer decomposable architectures, they do not provide closed-form symbolic expressions nor a principled pipeline for transforming neural estimators into interpretable representations; second, the method proposed by Chen et al. (2025) is suitable only for ATE prediction, and third, our pipeline regularizes the functional behavior learned by neural networks, which can exhibit spiking or irregular patterns, while the simplification steps of our pipelie produce smoother, more stable mappings, which directly enhances interpretability and faithfulness in some real world problems (Liu et al., 2024b; Wang et al., 2025)

## 3 PRELIMINARIES

We consider an observational dataset $\mathcal{D} = \{\boldsymbol{x}_i, t_i, y_i\}_{i=1}^N$ of $N$ i.i.d. samples from an unknown distribution $P(\mathbf{x}, \mathbf{t}, \mathbf{y})$, where $\mathbf{x} \in \mathcal{X} \subset \mathbb{R}^D$ are covariates, $\mathbf{t} \in \mathcal{T} = \{0, 1\}$[1] is the treatment, and $\mathbf{y} \in \mathbb{R}$ is the outcome. Our goal is to estimate the causal effect of the treatment on the outcome and provide a closed-form, interpretable formula for it.

Following the Neyman–Rubin potential outcomes framework (Rubin, 1974; Splawa-Neyman et al., 1990), each individual $i$ has potential outcomes $\mathbf{y}_i(t)$ for $t \in \{0, 1\}$. Only the outcome corresponding to the received treatment is observed, which constitutes the *fundamental problem of causal inference*. The target estimand is the individual treatment effect (ITE): $\tau_i = \mathbf{y}_i(1) - \mathbf{y}_i(0)$.

Since $\tau_i$ is not observable, we estimate instead the conditional average treatment effect (CATE):

$$\tau(\boldsymbol{x}) := \mathbb{E}[\mathbf{y}(1) \mid \mathbf{x} = \boldsymbol{x}] - \mathbb{E}[\mathbf{y}(0) \mid \mathbf{x} = \boldsymbol{x}], \quad \text{denoting} \quad \mu_t(\boldsymbol{x}) := \mathbb{E}[\mathbf{y}(\mathbf{t}) \mid \mathbf{x} = \boldsymbol{x}]. \quad (1)$$

We assume the standard conditions of causal inference: **i)** *positivity*, $0 < P(\mathbf{t} = t \mid \mathbf{x}) < 1$; **ii)** *conditional ignorability*, $\mathbf{y}(t) \perp\!\!\!\perp \mathbf{t} \mid \mathbf{x}$, which requires a valid adjustment set blocking all backdoor paths[2]; **iii)** *consistency*, $\mathbf{y}_i(t_i) = y_i$; and **iv)** *no interference*, $\mathbf{y}_i \perp\!\!\!\perp \mathbf{y}_j$ for $i \neq j$. Under these assumptions, it follows that $\mathbb{E}[\mathbf{y} \mid \mathbf{x}, \mathbf{t}]$ equals $\mu_t(\mathbf{x})$ (Hernán and Robins, 2025). The challenge is that this conditional expectation becomes increasingly difficult to estimate with high-dimensional, multimodal covariates and complex covariate–treatment–outcome relations, motivating the use of flexible function approximators such as neural networks.

---

[1]We focus on binary treatments for clarity, though the derivations extend to multi-valued and in some cases continuous treatments; see App. A.1.

[2]An adjustment set must block all backdoor paths between treatment and outcome, avoid selection bias, and not block front-door paths.

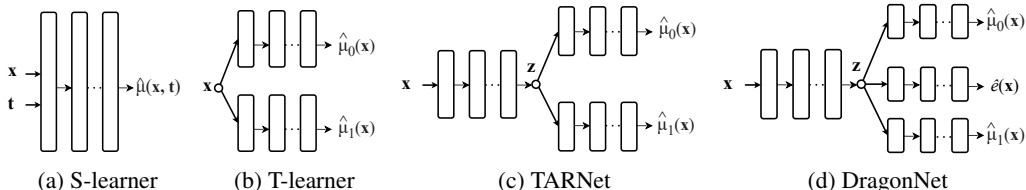

(a) S-learner      (b) T-learner      (c) TARNet      (d) DragonNet

Figure 3: Architectures used for *potential outcome regression*. Boxes denote layers or backbones (neural networks or KANs); dotted arrows indicate optional hidden layers.

## 3.1 CAUSAL NEURAL NETWORKS

Neural networks are flexible function approximators (Hornik et al., 1989) and often provide state-of-the-art potential–outcome and CATE estimates (Alaa and Schaar, 2018; Curth and Van der Schaar, 2021b; Tesei et al., 2023). We summarize the canonical architectures used in our experiments (see Fig. 3); all perform potential–outcome regression, $\mu_t(\boldsymbol{x})$, and obtain the CATE by difference, $\hat{\tau}(\boldsymbol{x}) = \hat{\mu}_1(\boldsymbol{x}) - \hat{\mu}_0(\boldsymbol{x})$. Our replacement of neural backbones by KANs is orthogonal to these designs and can also be applied to methods that target CATE directly (e.g., X-, R-learners, MRIV-Net; Frauen and Feuerriegel, 2022; Künzel et al., 2019; Nie and Wager, 2021).

**Meta-learners** (Künzel et al., 2019) are model-agnostic. The *S-learner* fits a single regressor $\mu_t(\boldsymbol{x}, t)$ of the factual outcome and obtains potential outcomes by toggling $t$. Its simplicity and ability to handle continuous treatments are appealing, but it may underuse $t$ when the treatment signal is weak relative to $\boldsymbol{x}$. The *T-learner* trains two separate regressors, $\hat{\mu}_0(\boldsymbol{x})$ and $\hat{\mu}_1(\boldsymbol{x})$, one per arm. This allows treatment-specific fits but splits the data, which can increase variance and yield sharp CATE estimates in small samples.

**TARNet** (Shalit et al., 2017) introduces a shared representation $\mathbf{z}(\boldsymbol{x})$ feeding two heads, combining the strengths above: shared structure across arms (mitigating data inefficiency) with treatment-specific outcome mappings (reducing treatment ignorance). Its CFRNet variant changes only the loss to encourage balanced representations, while the architecture remains identical. TARNet has shown robust performance and is widely used as a baseline (Curth and Van der Schaar, 2021a;b; Schwab et al., 2018; Shalit et al., 2017; Yao et al., 2018).

**DragonNet** (Shi et al., 2019) extends TARNet with a third head for the propensity score, $\hat{e}(\boldsymbol{x})$, encouraging $\mathbf{z}(\boldsymbol{x})$ to capture confounding structure through multitask learning. Originally paired with targeted regularization for doubly robust average treatment effect (ATE) estimation (Van der Laan et al., 2011), the base network already yields accurate potential outcomes and thus CATEs (Curth and Van der Schaar, 2021b; Ling et al., 2023; Lolak et al., 2025). Intuitively, predicting both outcomes and treatment forces the representation to retain covariates that co-determine assignment and response, which benefits counterfactual prediction.

Fig. 3 illustrates the corresponding schematics; we keep the architectures unchanged when replacing neural components by KANs.

## 3.2 KOLMOGOROV–ARNOLD NETWORKS

We employ KANs as our backbones. Kolmogorov–Arnold Networks (KANs) (Liu et al., 2024b) are deep models that replace fixed node-wise activations with *learnable univariate functions on edges*, using *addition*—and, in extensions, multiplication (Liu et al., 2024a)—as the only explicit multivariate operations. Their design is motivated by the Kolmogorov–Arnold representation theorem (KART), which guarantees that any continuous $f : \mathbb{R}^D \to \mathbb{R}$ can be expressed as

$$f(\boldsymbol{x}) = \sum_{q=1}^{2D+1} \Phi_q\Big( \sum_{k=1}^{D} \phi_{q,k}(x_k) \Big), \tag{2}$$

where all nonlinearities are one-dimensional (Arnold, 1957; Braun and Griebel, 2009; Kolmogorov, 1956; 1957). This motivates stacked architectures where multivariate structure emerges from compositions of simple univariate transformations and aggregations. Formally, a depth-$L$ KAN with widths

$\{n_l\}$ and coordinates $z_{l,r}$ replaces the linear map and fixed activation at layer $l$ by

$$z_{l+1,s} \;=\; \sum_{r=1}^{n_l} \phi_{l,s,r}\big(z_{l,r}\big), \quad s = 1, \ldots, n_{l+1}, \tag{3}$$

so that the whole network computes: $\text{KAN}(\boldsymbol{x}) \;=\; \big(\boldsymbol{\Phi}_{L-1} \circ \boldsymbol{\Phi}_{L-2} \circ \cdots \circ \boldsymbol{\Phi}_0\big)(\boldsymbol{x})$. Each $\phi_{l,s,r}$ is parameterized as a smooth B-spline,

$$\phi(z) \;=\; w_b\, b(z) + w_s\, \text{spline}(z),$$

with $b(z)$ a fixed baseline (e.g., identity or SiLU), and $w_s, w_b$ are learnable weights. This construction is fully differentiable and trainable with standard backpropagation. While the shallow KART decomposition may involve irregular univariates, stacking layers yields smooth, accurate approximations (Liu et al., 2024b).

KANs retain universal approximation: refining spline grids (internal degrees of freedom) and stacking layers (external degrees of freedom) expands expressivity while exposing interpretable one-dimensional components. Compared to MLPs with fixed nonlinearities, complexity shifts from dense weight matrices to a small number of spline coefficients per edge, making the active graph (which edges matter) separable from the functional form of each transformation. Training follows standard optimizers and losses, with sparsity encouraged by $\ell_1$ or group penalties that prune low-contribution edges. The pruned networks are compact, with remaining splines straightforward to inspect or approximate symbolically. The *MultKAN* extension (Liu et al., 2024a) further augments summation nodes with parameter-free multiplication nodes, explicitly representing interactions without forcing splines to emulate them.

In summary, KANs are deep models built from learnable one-dimensional splines composed through sums (and optionally products). They preserve universal approximation, train with off-the-shelf methods, admit effective pruning, and expose interpretable building blocks at the level of univariate functions and explicit interactions. These properties make them natural candidates to replace neural backbones in causalKANs.

## 4 CAUSAL KOLMOGOROV–ARNOLD NETWORKS

We introduce *causalKANs*, a framework that replaces the neural components of standard potential–outcome regressors with Kolmogorov–Arnold Networks (KANs), and augments training with *pruning* and *auto–symbolic search* to yield analytic, interpretable CATE formulas. The approach is model–agnostic: any causalNN can be converted by swapping MLP blocks with KAN (or MultKAN) blocks while keeping inputs and outputs unchanged, and using the same loss functions as their neural counterparts but augmented with regularization terms.

### 4.1 INTERPRETABLE CAUSAL LEARNING

Our goal is to deliver *closed-form* and *auditable* estimates of $\hat{\tau}(\boldsymbol{x}) = \hat{\mu}_1(\boldsymbol{x}) - \hat{\mu}_0(\boldsymbol{x})$ without sacrificing the flexibility of deep learning. To this end, we expose a pipeline in which the practitioner controls each simplification step, chooses which steps to perform, and sets explicit performance budgets that cap the error introduced by simplification. The five stages are shown in Fig. 19; architecture substitution is illustrated in Fig. 1, and implementation details are in App. B.

1. **KAN-ification (architecture swap).** Choose a causalNN (e.g., S-KAN , T-KAN , TARKAN , DragonKAN ) and replace each MLP subnetwork by a KAN block with matching input/output dimensions. When the original network has multiple subnetworks, intermediate widths and representation sizes need not be preserved; KAN widths can be reduced to encourage parsimony. MultKAN nodes are optional and only used when explicit interactions are needed (§3.2).

2. **Hyperparameter selection & training.** Tune depth, width, spline grid size, and regularization. During training we employ: **i)** *edge activity* penalty ($\ell_1$) $\lambda_1 \sum_{l,s,r} \mathbb{E}\big|\phi_{l,s,r}(z_{l,r})\big|$ to promote sparse parent sets; **ii)** *spline coefficient* regularization $\lambda_c \sum |w_s| + \lambda_s \sum |w_s - w_s'|$ to shrink magnitudes and encourage smoothness; and **iii)** an *entropy* penalty on fan-in/fan-out to discourage diffuse connectivity. These terms stabilize pruning and symbolification. Early stopping is applied

on a validation split. For notational brevity, all penalties are aggregated into $\mathcal{R}$ in later formulas. Among models with statistically indistinguishable validation metrics, we select the simplest (fewer layers, coarser grids, fewer nodes, no multiplication nodes), which is supported by an ablation study in App. C.2. We also compare the predictive loss (excluding $\mathcal{R}$) to the original causalNN; if the causalKAN loss exceeds the baseline by more than a user budget $\Lambda_{\text{arch}}$, we revise the KAN design.

3. **Pruning (structure simplification).** On a held-out set, compute edge importance scores

$$s_{l,s,r} \; = \; \mathbb{E}\big[\, \big|\phi_{l,s,r}(z_{l,r})\big|\,\big], \tag{4}$$

remove edges (and isolated nodes) with $s_{l,s,r} < \Gamma$, and retrain briefly if desired. There are alternative pruning techniques, that remove edges based on other metrics, see (Liu et al., 2024a). This step is optional: if the held-out loss rises beyond a pruning budget $\Lambda_{\text{prune}}$, we revert the change. Pruning exposes smaller parent sets and simplifies the subsequent symbolification.

4. **Auto–symbolic search (function simplification).** For each remaining edge function, fit a simple atom from a dictionary (identity, polynomials, $\tanh$, $\sin$, $\cos$, $\log$, $\exp$, etc.) via

$$\min_{m,a,b,c,d} \; \frac{1}{|\mathcal{V}|} \sum_{z \in \mathcal{V}} \Big(\phi_{l,s,r}(z) - \big[c\, f_m(az+b) + d\big]\Big)^2. \tag{5}$$

We introduce an interpretability-first inductive bias: attempt the simplest atoms first (polynomials); if the $R^2$ exceeds a threshold $\Gamma_{R^2}$, accept immediately without testing more complex atoms. Otherwise, continue through the dictionary and accept a substitution only if the validation loss increases by at most $\Lambda_{\text{symb}}$. If the increase exceeds the budget, revert. This design makes the loss–simplicity trade-off explicit and user-controllable.

5. **CATE extraction and interpretation.** Compose the simplified univariate functions to obtain closed-form heads $\hat{\mu}_0(\boldsymbol{x})$ and $\hat{\mu}_1(\boldsymbol{x})$ and thus an explicit $\hat{\tau}(\boldsymbol{x})$, which we also simplify algebraically. These expressions are executable and auditable, and they can be inspected term by term or transported across settings by substituting domain-grounded functions if needed.

At stages (3) and (4) we enforce an accept–reject gate: a structural or symbolic change is kept only if the held-out performance stays within its budget ($\Lambda_{\text{prune}}$, $\Lambda_{\text{symb}}$); otherwise it is reverted. Practitioners may also choose to *skip* pruning and/or symbolification entirely, yielding a continuum from fully flexible KANs (budgets set to 0) to sparse, fully symbolified formulas (larger budgets). This is crucial in regulated or high-stakes settings: one can freeze the pipeline at the desired interpretability level and document any accuracy impact.

When $L = 1$ and no MultKAN nodes are used, each head reduces to a Kolmogorov–Arnold Additive Model (KAAM). In this setting, we provide **i)** *probability radar plots* (PRPs) summarizing each variable's contribution relative to the average outcome, and **ii)** *partial dependence plots* (PDPs) showing the variation of the outcome as a function of a single covariate (Almodóvar et al., 2025; Knottenbelt et al., 2024). For deeper KANs or any use of MultKAN nodes, we currently provide only the closed-form expressions; visualization tools for these more complex models are left as future work. A detailed description and examples for KAAM-based plots appear in App. C.3.

In summary, the pipeline makes the accuracy–interpretability trade-off *controlled and reproducible*. Budgets ($\Lambda_{\text{arch}}, \Lambda_{\text{prune}}, \Lambda_{\text{symb}}$) bound the deviation from the original predictive performance; every accepted change is logged, and every rejected change is reverted. The procedure is architecture-agnostic, produces executable closed-form $\hat{\mu}_0(\boldsymbol{x})$, $\hat{\mu}_1(\boldsymbol{x})$, and $\hat{\tau}(\boldsymbol{x})$, and can be halted at any stage depending on the practitioner's needs. We adopt this protocol across all causalKANs variants and experiments that follow.

Beyond the universal approximation, KANs are most useful when their inductive biases match the data generating process, in particular when response surfaces are smooth and approximately decomposable into low dimensional additive components. This is consistent with evidence from physics informed and PDE benchmarks, where KANs outperform MLPs under smooth, structured dynamics (Wang et al., 2025), and with the original KAN work, which argues that many real world systems admit sparse and smooth functional structure (Liu et al., 2024b). From a causal perspective, treatment effect functions are often simpler than the underlying potential outcomes (Curth and Van der Schaar, 2021a), so causalKANs can leverage pruning and symbolic substitution as implicit regularizers that bias

toward simple, interpretable effect formulas, in contrast to standard MLP backbones. This behavior is illustrated in our SCM experiment (Section 5.3), where symbolic KANs recover the correct smooth additive structure and outperform both MLPs and non symbolic KANs, and suggests that causalKANs are particularly suitable in domains where smoothness and approximate additivity are plausible, such as physics based models, biomedical dose–response, or structured policy applications.

## 4.2 CausalKANs variants

We instantiate four canonical architectures, standard baselines in the CATE literature, to enable fair comparisons with prior work. These are representative examples: the same substitution process applies to other causalNNs, see the App. B.1 for an extended discussion. Fig. 3 shows a schematic view where each block can be a KAN, and App. B details specific implementations.

**S-KAN** (*causalKAN for S-Learner*) uses a single KAN to predict outcomes:

$$\hat{\mu}(\boldsymbol{x}, t) = KAN(\boldsymbol{x}, t), \quad \hat{\mu}_0(\boldsymbol{x}) = \hat{\mu}(\boldsymbol{x}, 0), \quad \hat{\mu}_1(\boldsymbol{x}) = \hat{\mu}(\boldsymbol{x}, 1). \tag{6}$$

With one KAN layer, Eq. 6 reduces to an additive model (KAAM). S-KAN naturally supports continuous treatments, since $\mu_t(\boldsymbol{x}, \mathbf{t})$ can be evaluated for $\mathbf{t} \in \mathbb{R}$.

**T-KAN** (*causalKAN for T-Learner*) employs two independent KAN heads:

$$\hat{\mu}_0(\boldsymbol{x}) = KAN_0(\boldsymbol{x}), \qquad \hat{\mu}_1(\boldsymbol{x}) = KAN_1(\boldsymbol{x}). \tag{7}$$

Each head is updated only with its respective treatment group, using

$$\mathcal{L}_{\text{T-KAN}} = \frac{1}{N} \sum_{i=1}^{N} \left[ t_i \big(y_i - \hat{\mu}_1(\boldsymbol{x}_i)\big)^2 + (1 - t_i)\big(y_i - \hat{\mu}_0(\boldsymbol{x}_i)\big)^2 \right] + \mathcal{R}. \tag{8}$$

This setup allows treatment-specific fits but may increase variance due to data splitting. Either head can be restricted to KAAM for maximal simplicity.

**TARKAN** (*causalKAN for TARNet*) first computes a representation vector,

$$\mathbf{z}(\boldsymbol{x}) = KAN_z(\boldsymbol{x}) \in \mathbb{R}^{D_{\mathbf{z}}}, \tag{9}$$

then feeds it to two KAN heads as in T-KAN :

$$\hat{\mu}_0(\boldsymbol{x}) = KAN_0(\mathbf{z}(\boldsymbol{x})), \quad \hat{\mu}_1(\boldsymbol{x}) = KAN_1(\mathbf{z}(\boldsymbol{x})). \tag{10}$$

Training follows the T-style loss Eq. 8, applied after the representation. This design parallels TARNet and relates to Mehendale et al. (2025), though we do not impose explicit distribution-matching on $\mathbf{z}$.

**DragonKAN** (*causalKAN for DragonNet*) augments TARKAN with a propensity head:

$$\hat{e}(\boldsymbol{x}) = \sigma\big(KAN_e(\mathbf{z}(\boldsymbol{x}))\big), \tag{11}$$

where $\sigma$ is the sigmoid function. Training adds a cross-entropy penalty on $\hat{e}(\boldsymbol{x})$ to the outcome losses in Eq. 8, encouraging $\mathbf{z}$ to capture confounding. We omit targeted regularization layers to preserve formula simplicity, focusing on potential–outcome regression and CATE estimation. Unlike S-KAN , these architectures (T-KAN, TARKAN, DragonKAN) are limited to binary or discrete treatments.

## 5 Experimental evaluation

We evaluate *causalKANs* on semi-synthetic benchmarks where ground-truth potential outcomes are available: ACIC 2016 (Dorie et al., 2019) and IHDP (Hill, 2011), settings A and B. Semi-synthetic data allow objective assessment of CATE accuracy despite the fundamental problem of causal inference. We report **i)** the Precision in Estimation of Heterogeneous Effect (PEHE) and **ii)** ATE error. Given test set $\mathcal{D}_{\text{test}}$, PEHE is $\sqrt{\frac{1}{|\mathcal{D}_{\text{test}}|} \sum \big(\hat{\tau}(x_i) - \tau(x_i)\big)^2}$, and ATE error is $|\hat{\tau} - \tau|$, where $(\hat{\tau}, \tau)$ are the estimated and the real ATE, computed as $\tau = \frac{1}{|\mathcal{D}_{\text{test}}|} \sum \tau(x)$. All training details follow §4.

**Datasets.** *IHDP (A/B)*. The IHDP benchmark (Hill, 2011) is constructed from covariates of a real observational study, with synthetic outcomes generated from known functions. We use the standard

100 replications for settings A and B, where A corresponds to a linear outcome surface and B introduces nonlinear and heterogeneous effects. *ACIC 2016.* The ACIC'16 challenge datasets (Dorie et al., 2019) provide covariates from administrative health records and semi-synthetic treatments and outcomes generated through nonlinear mechanisms. We use the nonlinear regimes commonly adopted in prior work. *NSLM.* The NSLM benchmark uses covariates from the National Study of Learning Mindset (Yeager et al., 2019), with a semi-synthetic DGP introduced by Carvalho et al. (2019). *NEWS.* The NEWS dataset contains text-derived covariates from a corpus of 5000 documents (Johansson et al., 2016). We adopt the DGP proposed by Crabbé et al. (2022). *TCGA.* The TCGA benchmark uses RNA-seq covariates from the Cancer Genome Atlas (Weinstein et al., 2013), combined with the semi-synthetic DGP of Zhang et al. (2023). Further details for all datasets are provided in App. C.1.

## 5.1 COMPARISON WITH NEURAL NETS

**Baselines.** We benchmark S-/T-KAN, TARKAN, and DragonKAN against their MLP-based counterparts (S-/T-learner, TAR-Net, DragonNet). We intentionally restrict baselines to neural counterparts to test our central claim—*comparable accuracy with improved interpretability*—rather than to chase leaderboards.

We trained causalNNs with the same hyperparameter budget as causalKANs, modifying depth, number of neurons, activations, regularization and learning rate, and selected the best model based on validation loss (we justify this choice in App. C.2).

Tab 1 reports PEHE and ATE error per dataset/setting. We assess statistical significance using the Friedman test with Holm-corrected Wilcoxon signed-rank post-hoc comparisons at $\alpha = 0.05$ (see Demšar (2006); Rainio et al. (2024) for a details on these tests). In the tables, the best-performing model (baseline) is underlined. Models in **bold** are those for which we cannot reject the null hypothesis of equal performance with respect to the baseline ($p \geq 0.05$). A train/val/test split of $80/10/10$ was leveraged, as well as early stopping and Adam optimizer (Kingma and Ba, 2014).

We observe from both Tab 1 and Fig. 2 that, in 7 of the 8 datasets, there exists at least one instance of causalKAN that is the best model or whose performance is statistically indistinguishable from the best-performing neural model, in the sense that we fail to reject the null hypothesis

Table 1: **Out-of-sample** ATE error and PEHE for KAN and MLP across datasets. The baseline (best) according to the Friedman corrected test is underlined, and all models not statistically different are **bolded** ($\alpha >= 0.05$ in a paired Wilcoxon corrected test). Values are reported as mean$_{std}$.

| Dataset | Model | KAN | | MLP | |
|---|---|---|---|---|---|
| | | ATE err | PEHE | ATE err | PEHE |
| IHDP A | S-Learner | **$0.19_{0.35}$** | **$0.98_{1.03}$** | **$0.23_{0.47}$** | $1.04_{1.30}$ |
| | T-Learner | **$0.13_{0.08}$** | **$\underline{0.62_{0.28}}$** | $0.24_{0.27}$ | $0.90_{0.51}$ |
| | TarNet | **$0.13_{0.10}$** | **$0.64_{0.31}$** | $0.17_{0.27}$ | **$0.70_{0.78}$** |
| | DragonNet | $\underline{0.14_{0.10}}$ | **$0.66_{0.35}$** | $0.17_{0.27}$ | **$0.68_{0.77}$** |
| IHDP B | S-Learner | $0.37_{0.37}$ | $3.01_{0.63}$ | $0.32_{0.24}$ | $2.63_{0.58}$ |
| | T-Learner | $0.34_{0.27}$ | $2.80_{0.44}$ | **$0.25_{0.20}$** | **$2.06_{0.35}$** |
| | TarNet | $0.33_{0.24}$ | $2.68_{0.44}$ | **$\underline{0.23_{0.20}}$** | **$2.08_{0.36}$** |
| | DragonNet | **$0.28_{0.22}$** | **$2.64_{0.44}$** | $\underline{0.25_{0.21}}$ | $\underline{2.00_{0.34}}$ |
| ACIC 2 | S-Learner | **$0.20_{0.38}$** | **$0.20_{0.38}$** | $0.38_{0.44}$ | $0.74_{0.38}$ |
| | T-Learner | $0.56_{0.42}$ | $1.43_{0.72}$ | $1.97_{2.13}$ | $7.05_{5.48}$ |
| | TarNet | $0.25_{0.33}$ | $0.78_{0.44}$ | $0.36_{0.44}$ | $0.96_{0.83}$ |
| | DragonNet | **$0.21_{0.26}$** | $0.75_{0.33}$ | $0.36_{0.44}$ | $0.96_{0.82}$ |
| ACIC 7 | S-Learner | $0.66_{0.75}$ | $4.13_{1.56}$ | $0.75_{0.60}$ | $4.51_{1.54}$ |
| | T-Learner | **$0.43_{0.43}$** | **$3.06_{1.18}$** | $1.37_{2.14}$ | $7.05_{6.64}$ |
| | TarNet | **$\underline{0.42_{0.43}}$** | **$3.06_{1.18}$** | $0.58_{0.50}$ | $4.17_{1.47}$ |
| | DragonNet | **$0.43_{0.43}$** | **$\underline{3.06_{1.17}}$** | $0.59_{0.50}$ | $4.17_{1.48}$ |
| ACIC 26 | S-Learner | **$0.42_{0.45}$** | $3.23_{1.57}$ | $0.75_{0.60}$ | $4.51_{1.54}$ |
| | T-Learner | **$0.36_{0.34}$** | **$2.80_{1.08}$** | $1.37_{2.14}$ | $7.05_{6.64}$ |
| | TarNet | **$\underline{0.35_{0.34}}$** | **$2.80_{1.08}$** | $0.58_{0.50}$ | $4.17_{1.47}$ |
| | DragonNet | **$0.35_{0.34}$** | **$\underline{2.79_{1.08}}$** | $0.59_{0.50}$ | $4.17_{1.48}$ |
| NSLM | S-Learner | **$0.048_{0.038}$** | **$0.752_{0.008}$** | $0.190_{0.032}$ | $0.753_{0.006}$ |
| | T-Learner | **$0.050_{0.033}$** | **$0.752_{0.006}$** | **$0.048_{0.033}$** | $0.756_{0.007}$ |
| | TarNet | $0.055_{0.034}$ | **$\underline{0.752_{0.006}}$** | **$0.049_{0.033}$** | $0.755_{0.007}$ |
| | DragonNet | $0.058_{0.040}$ | $\underline{0.753_{0.007}}$ | **$0.050_{0.035}$** | $0.756_{0.007}$ |
| NEWS | S-Learner | **$2.01_{0.68}$** | **$0.141_{0.119}$** | $2.73_{1.04}$ | $0.284_{0.311}$ |
| | T-Learner | **$2.04_{0.53}$** | $0.192_{0.136}$ | $2.12_{0.54}$ | $0.158_{0.126}$ |
| | TarNet | **$\underline{1.97_{0.48}}$** | $0.157_{0.125}$ | **$2.05_{0.44}$** | **$\underline{0.120_{0.112}}$** |
| | DragonNet | **$2.04_{0.72}$** | $0.176_{0.125}$ | $2.11_{0.44}$ | **$0.128_{0.112}$** |
| TCGA $\times 10^{-2}$ | S-Learner | $2.50_{1.50}$ | $4.64_{1.26}$ | $0.03_{0.02}$ | $2.18_{0.27}$ |
| | T-Learner | $1.98_{6.36}$ | $4.41_{6.27}$ | **$\underline{0.02_{0.01}}$** | $2.15_{0.07}$ |
| | TarNet | $1.19_{0.92}$ | $3.26_{0.65}$ | $0.04_{0.01}$ | **$\underline{1.81_{0.03}}$** |
| | DragonNet | $14.95_{33.92}$ | $20.34_{33.81}$ | $0.05_{0.04}$ | $3.74_{0.27}$ |

that the models achieve the same value of the evaluation metric. On the other hand, in the TCGA dataset, none of the causalKANs achieve competitive results. However, that does not prevent the use of our pipeline, since one of the steps of it is to compare the metrics against causalNNs. If metrics of causalKANs are not satisfactory, then the use of KAN based predictors is not recommended by our pipeline (see line 14 of Alg. 1).

## 5.2 INTERPRETABILITY RESULTS

We show here some examples that highlight the interpretability of causalKANs, and we will refer to App. C.3 for more examples and details on these representations. We selected randomly one realization of the shown datasets.

**ACIC-7.** We show results for a single layer T-KAN (as it is an additive model, we call it T-KAAM) for ACIC-7, which has been demonstrated very high-performance according to Tab 1. This model yields a function of *heterogeneous* CATE, which is a generalized additive model of the covariates (see App. B). In addition to the closed-form CATE, we provide in Fig. 4 (extended in App. C.3.1) a *Radar plot (Left)* that compares for two individuals, what is the contribution of each variable to the CATE, compared

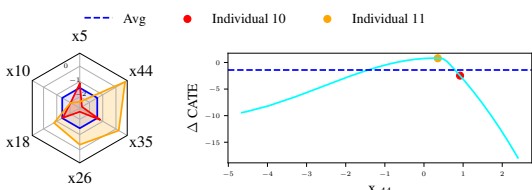

Figure 4: Radar plot and PDP for variable contribution to CATE, using T-KAAM in ACIC-7.

with the average outcome, and a *PDP (Right)* that represent the curve that defines the variation of the CATE ($\Delta$ CATE) with a specific feature. In this case, we can observe how $x_{44}$ increases the CATE w.r.t. the average in individual 10, while it decreases the CATE in individual 11, because the shape of the curve that relates $x_{44}$ and CATE has a maximum near the value of $x_{44}$ of individual 10.

**IHDP A.** On the other hand, we can estimate *homogeneous* CATEs with a shallow S-KAN (S-KAAM), which provides closed-form homogeneous treatment effect. In IHDP A, where the known (denoised) ITE is 4 for every individual, we leverage S-KAAM to obtain the value of the causal effect. In Fig. 5 we can observe that the predicted formula has an additive linear term relative to the treatment, which correspond to the CATE (as developed in App. B). The contribution of other variables (not causal) can be consulted in App. C.3.1. In this case, the CATE can directly be consulted in the formula: 3.74 for the given example.

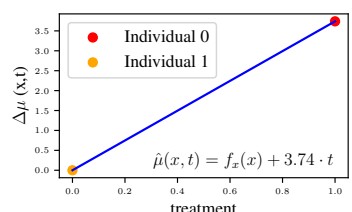

Figure 5: PDP for treatment contribution in $\hat{\mu}(\mathbf{x}, \mathbf{t})$ estimation, using S-KAAM in IHDP-A.

### 5.3 Symbolic CATE recovery

We have conducted two experiments with known structural causal models, to determine empirically how well causalKANs capture the equations of the data generating process (DGP). They also provide examples of when the regularization and the simplification steps of the pipeline of causalKANs help to achieve better approximations to the true data generating functions. Metrics can be consulted in App. C.4

In Fig. 6a, the function that generates the potential outcomes is additive in the treatment. Therefore, it lies in the class that S-KAAM can model (see App. B). It can be observed that **i)** the S-KAAM provides a function that is closer to the groundtruth compared with MLP and **ii)** the symbolic substitution step approximates the grountruth even better. Note that there is a cosntant value that S-KAAM cannot recover, but does not modify the CATE estimation, since each difference $\hat{f}(\mathbf{x}, t_1) - \hat{f}(\mathbf{x}, t_1)$ will cancel this constant.

On the other hand, Fig. 6b represents the approximation to the CATE with a binary treatment, as the difference between the two potential outcomes, $\{\mathbf{y}(1), \mathbf{y}(0)\}$. In this case, individual treatment effect is not constant across individuals, and the treatment does not contribute additively to the outcomes. However, the contribution to each variable to the ITE is additive. Therefore, this DGP lies in the class that T-KAAM can model (see App. B). In the same fashion as in the previous experiment, symbolic T-KAAM is the model that captures better the true function that generates the CATE.

## 6 Final remarks

We have presented causalKANs, a framework that transforms high-performing neural CATE estimators into knowledge-augmented networks with closed-form, auditable effect functions. On standard benchmarks, at least one causalKAN variant matches or surpasses its neural counterpart on PEHE/ATE, while shallow heads (KAAMs) often yield the most favorable accuracy-interpretability trade-off. The pruning and auto-symbolification stages expose explicit formulas and partial dependence plots (PDP; Friedman (2001)). These properties make causalKANs suitable for healthcare,

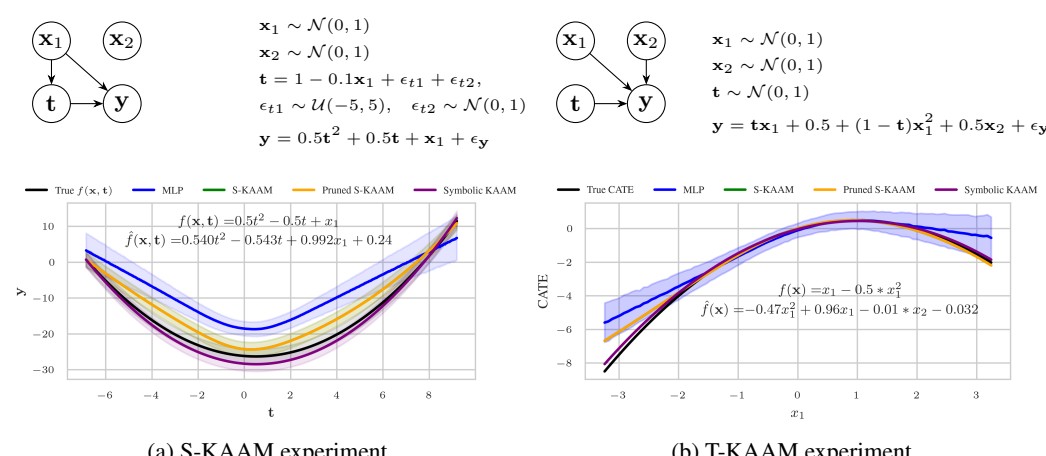

(a) S-KAAM experiment          (b) T-KAAM experiment

Figure 6: Synthetic SCMs and the curves recovered by causalKANs in each step of the pipeline. In blue, the curve recovered by MLP counterparts. Curves green and yellow are overlapped, because the effect of pruning is not appreciable. $f$ and $\hat{f}$ represent the groundtruth and the estimated function by Symbolic KAAM, respectively. All curves has been shifted so that the first point concides. *(Left)* represents directly the equation of the outcome and covariates provided by S-KAAM. *(Right)* represents the equation of the CATE given by T-KAAM, as the difference of two additive formulas.

policy evaluation, and economics, where interpretability is central to adoption and auditing (Amann et al., 2020; Doshi-Velez and Kim, 2017; Rudin, 2018; Tonekaboni et al., 2019). Closed-form expressions can be scrutinized, simplified, or aligned with domain-grounded terms to support external validity. We release code to ensure reproducibility and facilitate adoption.

Nevertheless, causalKANs also present open challenges. Visualization tools are currently most effective for shallow KAAMs, while scaling them to deeper or more complex networks remains an area for development. Training and inference are generally more demanding than for standard neural models, and performance can be sensitive to regularization choices or pruning/symbolification budgets, which may introduce approximation errors or variability. As with all observational CATE estimation, our conclusions depend on standard identification assumptions (e.g., unconfoundedness, overlap), and our evaluation relies mainly on semi-synthetic datasets, so further assessment on field data is needed. Finally, uncertainty quantification for symbolic outputs and broader support for continuous or multi-valued treatments are still limited.

Building on these observations, future work should expand theoretical and empirical aspects of causalKANs. Methodologically, interpretability should be assessed with task-grounded and user-study metrics (Tonekaboni et al., 2019), and contrasted with post-hoc explanations such as LIME and SHAP (Lundberg and Lee, 2017; Ribeiro et al., 2016, routinely used but with different guarantees). Developing principled surrogate metrics for model selection, and procedures that reduce accuracy gaps across the simplification stages, would improve stability and reliability. Extending causalKANs to continuous and multi-treatment settings, as well as deriving finite-sample guarantees for CATE estimation, remain important directions. Incorporating inverse-probability weighting, doubly robust objectives, and batch sampling techniques would further broaden scope. Empirically, validation on real-world data, ideally against interventin. Another avenue is to adapt causalKANs to non-tabular modalities such as images, graphs, and genomic or sequence data by leveraging recent KAN variants for convolutional and graph architectures (Bodner and coauthors, 2024; Bresson and coauthors, 2024; Cherednichenko and coauthors, 2025; Liu et al., 2024b). Designing benchmarks with ground-truth potential outcomes in these modalities is itself an open challenge, and we leave multimodal CATE benchmarking with causalKANs to future work. Related with visualization, beyond PDPs, implementing related plot families, such as accumulated local effects (Apley and Zhu, 2020) or individual conditional expectation (Goldstein et al., 2015), may also be informative when applied directly to the closed-form formulas. Finally, exploring robustness to adversarial perturbations, distribution shifts, and fairness constraints (Rudin, 2018), together with reducing the performance gaps between KAN-ification, pruning, and symbolification, will be important for reliable deployment.

## Reproducibility Statement

We have taken several steps to ensure the reproducibility of our results. All causalKAN variants are defined with explicit mathematical formulations in §4.2, and the interpretability pipeline is described in detail in §4.1, including pruning and auto–symbolic search criteria, budgets, and regularization terms. The assumptions and problem setup are clearly stated in §3, while implementation details such as hyperparameter choices, training protocols, and evaluation procedures are provided in §5 and the appendix. All experiments were run on publicly available benchmark datasets. To facilitate reproducibility and comparison with future work, we release the full source code, including scripts to reproduce all figures and tables. Together, these resources ensure that both the methodological contributions and empirical findings of this paper can be independently verified and extended by the community.

## Ethics Statement

This work does not raise direct ethical concerns of that nature. All datasets used are publicly available benchmarks, processed following documented procedures. The proposed framework aims to improve transparency and interpretability in causal machine learning, which is a key requirement for responsible deployment of AI systems in high-stakes domains (Amann et al., 2020; Rudin, 2018). By providing closed-form analytic formulas and explicit control over the trade-off between accuracy and interpretability, our approach is designed to facilitate auditing, external validation, and trustworthy adoption. As with all machine learning methods, potential biases or limitations in the datasets may propagate to the models, but the interpretability of our approach makes such issues easier to identify and mitigate. This work adheres to the ICLR code of ethics.

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

# Appendix

## A TREATMENT SPACE GENERALIZATIONS

Let us generalize the theory explained in §3 to the settings of multiple treatment and continuous treatment. CausalKANs can also be generalized following the same fashion.

### A.1 MULTIPLE DISCRETE TREATMENTS

We extend the setup in §3 to $\mathbf{t} \in \mathcal{T} = \{1, \ldots, K\}$. For each $k \in \{1, \ldots, K\}$, define the potential outcome $\mathbf{y}(k)$ and the potential outcome regressor

$$\mu_k(\boldsymbol{x}) := \mathbb{E}[\mathbf{y}(k) \mid \mathbf{x} = \boldsymbol{x}]. \tag{12}$$

Pairwise conditional average treatment effects are contrasts

$$\tau_{ba}(\boldsymbol{x}) := \mu_b(\boldsymbol{x}) - \mu_a(\boldsymbol{x}), \qquad a, b \in \{1, \ldots, K\}. \tag{13}$$

Assumptions generalize in the standard way: **i)** positivity, $P(\mathbf{t} = k \mid \mathbf{x} = \boldsymbol{x}) > 0$ for all $k$; **ii)** conditional ignorability, $\mathbf{y}(k) \perp\!\!\!\perp \mathbf{t} \mid \mathbf{x}$ for all $k$; **iii)** consistency, $\mathbf{y}_i(t_i) = y_i$; and **iv)** no interference. Let the propensity vector be $e_k(\boldsymbol{x}) = P(\mathbf{t} = k \mid \mathbf{x} = \boldsymbol{x})$, $\sum_k e_k(\boldsymbol{x}) = 1$.

**S-learner (unchanged).** Use a single regressor $\hat{\mu}(\boldsymbol{x}, t)$; predicted heads are $\hat{\mu}_k(\boldsymbol{x}) = \hat{\mu}(\boldsymbol{x}, k)$. Train with factual MSE:

$$\mathcal{L}_{\text{S-Learner}} = \frac{1}{N} \sum_{i=1}^{N} \left( \hat{\mu}(\boldsymbol{x}_i, t_i) - y_i \right)^2. \tag{14}$$

**T-learner ($K$ heads).** Instantiate $K$ heads $\hat{\mu}_k(\boldsymbol{x})$ and update only the factual head:

$$\mathcal{L}_{\text{T-learner}} = \frac{1}{N} \sum_{i=1}^{N} \left( y_i - \hat{\mu}_{t_i}(\boldsymbol{x}_i) \right)^2. \tag{15}$$

**TARNet ($K$ heads).** Use a shared representation $\mathbf{z}(\boldsymbol{x}) \in \mathbb{R}^{D_\mathbf{z}}$ and $K$ heads:

$$\mathbf{z}(\boldsymbol{x}), \qquad \hat{\mu}_k(\mathbf{z}(\boldsymbol{x})) = \text{Head}_k(\mathbf{z}(\boldsymbol{x})), \quad \mathcal{L}_{\text{TARNet}} = \frac{1}{N} \sum_i \left( y_i - \hat{\mu}_{t_i}(\mathbf{z}(\boldsymbol{x}_i)) \right)^2. \tag{16}$$

**DragonNet ($K+1$ heads).** Add a propensity head with softmax output:

$$\hat{e}_k(\mathbf{z}(\boldsymbol{x})) = \text{softmax}_k(\mathbf{z}(\boldsymbol{x})), \quad \mathcal{L}_{\text{DragonNet}} = \mathcal{L}_{\text{TARNet}} + \lambda_{PS} \left[ -\frac{1}{N} \sum_i \log \hat{e}_{t_i}(\mathbf{z}(\boldsymbol{x}_i)) \right]. \tag{17}$$

After training, pairwise CATEs follow from equation 13 with $\mu_k$ replaced by $\hat{\mu}_k$.

### A.2 CONTINUOUS TREATMENTS

Let $\mathbf{t} \in \mathcal{T} \subset \mathbb{R}$ and define the dose–response function

$$\mu(\boldsymbol{x}, t) := \mathbb{E}[\mathbf{y}(t) \mid \mathbf{x} = \boldsymbol{x}]. \tag{18}$$

For any reference level $t_0 \in \mathcal{T}$, the conditional effect of moving from $t_0$ to $t$ is

$$\tau(\boldsymbol{x}, t) := \mu(\boldsymbol{x}, t) - \mu(\boldsymbol{x}, t_0), \tag{19}$$

and local effects can be summarized by the marginal treatment effect $\partial\mu(\boldsymbol{x}, t)/\partial t$ if desired. The identifiability assumptions extend as: positivity w.r.t. the treatment density $p(t \mid \mathbf{x})$, conditional ignorability $\mathbf{y}(t) \perp\!\!\!\perp \mathbf{t} \mid \mathbf{x}$ for all $t \in \mathcal{T}$, consistency, and no interference.

**Applicable architecture.** Among the causalNNs above, only the S-learner directly accommodates continuous $t$:

$$\mathcal{L}_{\text{S-cont}} = \frac{1}{N} \sum_{i=1}^{N} \left( \hat{\mu}(\boldsymbol{x}_i, t_i) - y_i \right)^2, \tag{20}$$

with $\hat{\tau}(\boldsymbol{x}, t) = \hat{\mu}(\boldsymbol{x}, t) - \hat{\mu}(\boldsymbol{x}, t_0)$. $\hat{\tau}(\boldsymbol{x}, t)$ can also be provided direclty as a function of the treatment. T-learner/TARNet/DragonNet require finitely many treatment-specific heads and hence are not directly applicable without discretization of $\mathcal{T}$; we therefore recommend the use of the S-learner in continuous-treatment settings.

## B DETAILS ON CAUSALKANS VARIANTS

As explained in §4, the adaptation of causalNNs to causalKANs consists of changing the MLP backbones by KANs. In Fig. 7, we include examples of the proposed causalKANs, with an arbitrary number of layers and summation nodes.

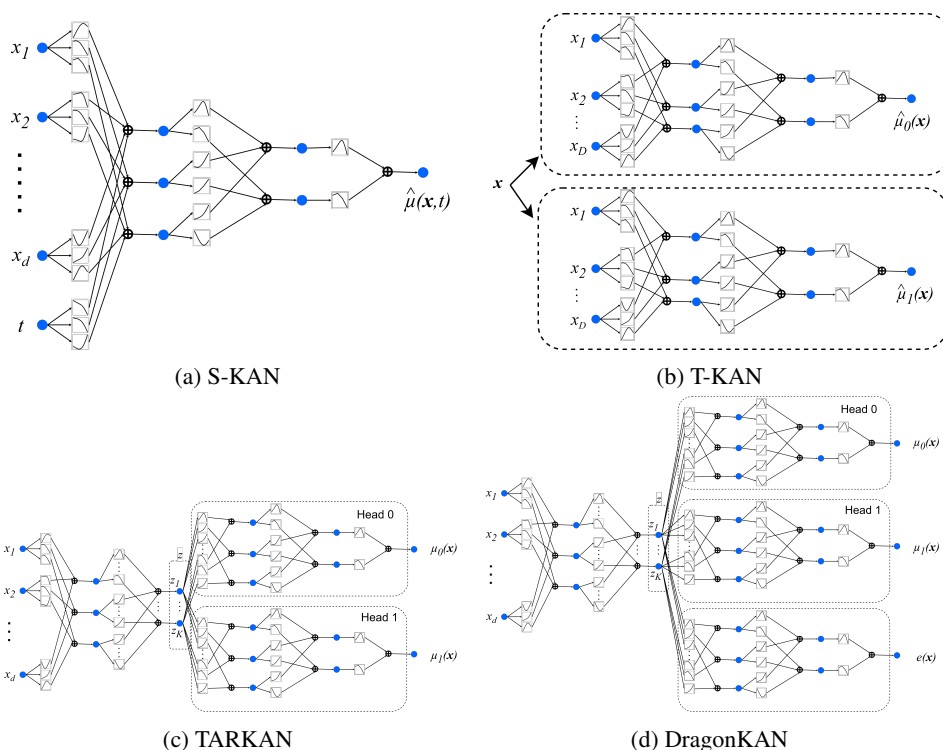

(a) S-KAN      (b) T-KAN

(c) TARKAN      (d) DragonKAN

Figure 7: CausalKANs detailed architectures. The number of layers of each model is arbitrary, just to show an example of each one.

From the point of view of interpretability, constructing very complex causalKANs is harmful for achieving simple and auditable formulas. Therefore, as mentioned in point 2 of §4.1, obtain the simplest model is crucial to improve interpretability, and we recommend to minimize the complexity of the network, among the models with similar performance.

We want to illustrate some interesting combinations of hyperparameters, that yield into special interpretable causal effects. In particular, when meta-learners employ additive models, we call them S-KAAM and T-KAAM, and when the heads of TARKAN and DragonKAN are also shallow additive models, we call them TARKAAM and DragonKAAM.

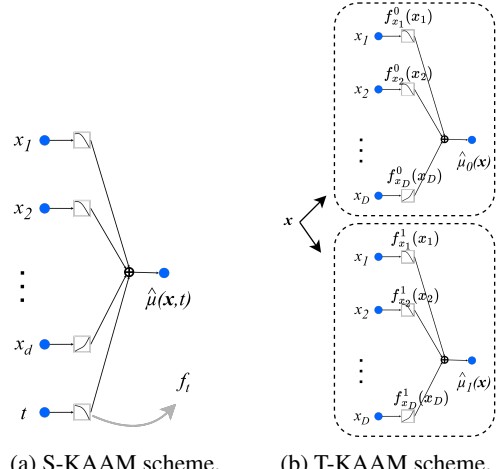

(a) S-KAAM scheme.  (b) T-KAAM scheme.

Figure 8: Particularly interpretable architectures of causalKANs. *(Left)* a single KAAM that predicts the outcome in an additive manner. *(Left)* two KAAMs, one for each treatment, which leads to an additive CATE function.

**S-KAAM.**  When training and S-KAN , if the hyperparameters selected results into a single-layer KAN, then we have a KAAM (Almodóvar et al., 2025). We call this model S-KAAM, and a scheme can be found in Fig. 8a. The interesting point is that, when we have an S-KAAM, the effects on the population are homogeneous, and we have access to the *effect curve* directly. The effect curve defines how the outcome varies with the treatment. Since, in a S-KAAM, the model is additive:

$$\hat{\mu}(\boldsymbol{x}, t) = f_{\boldsymbol{x}}(\boldsymbol{x}) + f_t(t), \tag{21}$$

we define the *effect curve*, $f_t : \mathcal{T} \to \mathbb{R}$, as the function that represents the variations of the outcome, depending on the treatment. Therefore, any causal effect of two different values of the treatment, $a, b$ can be computed as the difference along the effect curve:

$$\hat{\tau}_{ab}(\boldsymbol{x}) = \hat{\mu}(\boldsymbol{x}, b) - \hat{\mu}(\boldsymbol{x}, a) = [f_{\boldsymbol{x}}(\boldsymbol{x}) + f_t(b)] - [f_{\boldsymbol{x}}(\boldsymbol{x}) + f_t(a)] = f_t(b) - f_t(a), \tag{22}$$

Note that this model yields homogeneous treatment effects, making the CATE equivalent to the ATE, and removing the covariate dependence.

For a binary treatment, the CATE can be computed as:

$$\hat{\tau}(\boldsymbol{x}) = f_t(1) - f_t(0), \tag{23}$$

**T-KAAM.**  In the same fashion, if both subnetworks in a T-KAN are KAAMs, we can get a simple analytic solution of the causal effect, that depends on each covariate independently.

Having that each potential outcome can be expresed as:

$$\hat{\mu}_t(\boldsymbol{x}) = f_{x_1}^t(x_1) + f_{x_2}^t(x_2) + \cdots + f_{x_D}^t(x_D), \tag{24}$$

where $f_{x_i}^t$ is the activation function corresponding to the covariate $x_i$ in the subnetwork of the treatment $t$. We can compute the individual contribution of each variable, yielding in an estimation of the causal effect, for a binary treatment, as:

$$\hat{\tau}(\boldsymbol{x}) = \underbrace{[f_{x_1}^1(x_1) - f_{x_1}^0(x_1)]}_{\mathbf{g}_{x_1}(x_1)} + \underbrace{[f_{x_2}^1(x_2) - f_{x_2}^0(x_2)]}_{\mathbf{g}_{x_2}(x_2)} + \cdots + \underbrace{[f_{x_D}^1(x_D) - f_{x_D}^0(x_D)]}_{\mathbf{g}_{x_D}(x_D)} \tag{25}$$

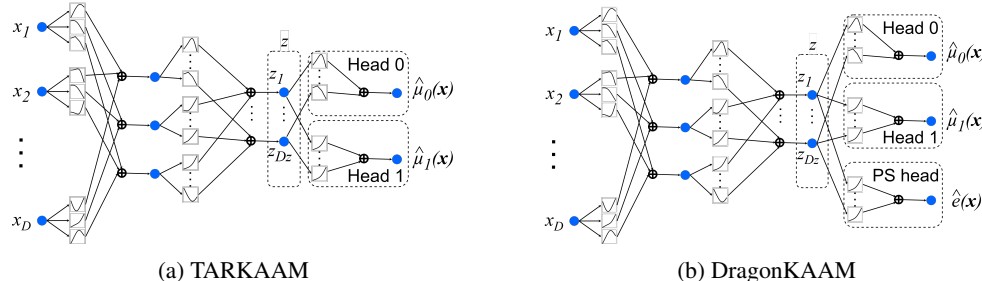

(a) TARKAAM          (b) DragonKAAM

Figure 9: TARKAN and DragonKAN with shallow heads. With respect to Fig. 7, the connections that depart from **z** have been reordered to make the equivalence more noticeable.

As an implementation detail, note that a T-KAAM can be implemented with a single KAN shallow model with two outputs, each one for $\hat{\mu}_1(\boldsymbol{x})$ and $\hat{\mu}_0(\boldsymbol{x})$, respectively.

**TARKAAN and DragonKAAM.** We would like to make an observation on the implementation of TARKAN and DragonKAN when the heads are shallow KAAMs. We can observe in Fig. 9 that, when instantiating a single KAN, with 2 and 3 output nodes, respectively. To construct those figures, we have reordered the last layer of a single KAN with 2/3 outputs, and show that is equivalent to add 2/3 KAAMs that have $\mathbf{z}(\boldsymbol{x})$ as input.

That is an interesting fact because we find, empirically, that TARKAAM and DragonKAAM are between the best-performer TAR-like and Dragon-like networks when evaluating in the IHDP and ACIC datasets. We also note that a KAN can learn identity splines, which could lead to have a representation vector equal to the input: $\mathbf{z}(\mathbf{x}) = \mathbf{x}$. If that is the case, the TAARKAM and DragonKAAM can be seen as T-KAAMs, since all the outputs are independent additive functions of the covariates (except for the common regularization term, $\mathcal{R}$). For example, that is the case for the dataset ACIC-7, which cause that T-KAN , TARKAN and DragonKAN to have the almost same metrics.

### B.1 EXTENDING CAUSALKANS TO OTHER ESTIMATORS

CausalKANs can be used as a drop in replacement in many causal estimators by KAN-ifying nuisance components (outcome regressions, propensity scores, CATE regressors) while leaving the estimand unchanged. After pruning and symbolification, these components become analytic expressions that can be inspected and manipulated. Which part should be made interpretable is estimator dependent and also dataset dependent, since different applications emphasize different aspects of the causal mechanism.

We consider binary treatment $\mathbf{t} \in \{0, 1\}$, outcome $\mathbf{y}$, covariates $\mathbf{x}$, propensity score $e(\mathbf{x}) = \mathbb{P}(\mathbf{t} = 1 \mid \mathbf{x})$, potential outcome regressions $\mu_0(\mathbf{x})$ and $\mu_1(\mathbf{x})$, and CATE $\tau(\mathbf{x}) = \mu_1(\mathbf{x}) - \mu_0(\mathbf{x})$ with estimator $\hat{\tau}(\mathbf{x})$.

**IPW.** Inverse probability weighting (IPW) estimates the ATE using only a propensity model $e(\mathbf{x})$ (Hernán and Robins, 2025; Rosenbaum and Rubin, 1983). For a sample $\{(\boldsymbol{x}_i, t_i, y_i)\}_{i=1}^n$,

$$\hat{\tau}_{\text{IPW}} = \frac{1}{n} \sum_{i=1}^n \left( \frac{t_i y_i}{\hat{e}(\boldsymbol{x}_i)} - \frac{(1 - t_i) y_i}{1 - \hat{e}(\boldsymbol{x}_i)} \right). \tag{26}$$

causalKANs enter through

$$\hat{e}(\mathbf{x}) = \sigma\big(f_{\text{KAN}}^{(e)}(\mathbf{x})\big), \tag{27}$$

so that the log odds

$$\log \frac{\hat{e}(\mathbf{x})}{1 - \hat{e}(\mathbf{x})} = f_{\text{KAN}}^{(e)}(\mathbf{x}) \tag{28}$$

is a closed form KAN expression. After pruning and symbolification, this gives an interpretable model of the selection mechanism (which covariates drive treatment assignment and how they interact), while $\hat{\tau}_{\text{IPW}}$ remains the same functional of the weights.

**X learner.** The X learner (Künzel et al., 2019) combines outcome regressions and effect regressions. Given $\hat{\mu}_0(\mathbf{x})$ and $\hat{\mu}_1(\mathbf{x})$, one constructs pseudo effects for treated and control units and learns two CATE regressors $\hat{\tau}(\mathbf{x})^{(1)}$ and $\hat{\tau}(\mathbf{x})^{(0}$, which are combined as

$$\hat{\tau}(\mathbf{x})_{\text{X}} = g(\mathbf{x})\,\hat{\tau}(\mathbf{x})^{(0)} + \big(1 - g(\mathbf{x})\big)\,\hat{\tau}(\mathbf{x})^{(1)}, \qquad (29)$$

with $g(\mathbf{x})$ often chosen as $\hat{e}(\mathbf{x})$.

CausalKANs can be used in two complementary ways:

- *KAN-ified potential outcomes:* set $\hat{\mu}_t(\mathbf{x}) = f_{\text{KAN}}^{(t)}(\mathbf{x})$ for $t \in \{0, 1\}$. After pruning and symbolification, $\hat{\mu}_0(\mathbf{x})$ and $\hat{\mu}_1(\mathbf{x})$ are analytic functions, so the practitioner can directly inspect how covariates shape each potential outcome and the induced pseudo effects.

- *KAN-ified CATE:* set $\hat{\tau}(\mathbf{x})^{(t)} = g_{\text{KAN}}^{(t)}(\mathbf{x})$ and optionally $g(\mathbf{x}) = \hat{e}(\mathbf{x}) = \sigma(f_{\text{KAN}}^{(e)}(\mathbf{x}))$. Then $\hat{\tau}(\mathbf{x})_{\text{X}}$ is an explicit combination of a small number of KAN terms, providing a closed form heterogeneous effect curve.

Which variant is preferable depends on whether interpretability should focus on the potential outcomes, the CATE, or the selection mechanism, and this is driven by the dataset and scientific question.

**AIPW, DR learners, and R learner.** Augmented IPW (AIPW) combines outcome and propensity models and is doubly robust (Bang and Robins, 2005; Robins et al., 1994; Tsiatis, 2006). A standard AIPW ATE estimator is

$$\hat{\tau}_{\text{AIPW}} = \frac{1}{n}\sum_{i=1}^{n}\left[\hat{\mu}_1(\boldsymbol{x}_i) - \hat{\mu}_0(\boldsymbol{x}_i) + \frac{t_i\big(y_i - \hat{\mu}_1(\boldsymbol{x}_i)\big)}{\hat{e}(\boldsymbol{x}_i)} - \frac{(1 - t_i)\big(y_i - \hat{\mu}_0(\boldsymbol{x}_i)\big)}{1 - \hat{e}(\boldsymbol{x}_i)}\right]. \qquad (30)$$

KAN-ifying $\hat{\mu}_0$, $\hat{\mu}_1$ and/or $\hat{e}$ yields interpretable models for baseline heterogeneity $\hat{\mu}_1(\mathbf{x}) - \hat{\mu}_0(\mathbf{x})$ and for the augmentation terms. For applications where the effect structure is primary, causalKANs on $\hat{\mu}_0$ and $\hat{\mu}_1$ are most informative; for settings where selection bias is central, causalKANs on $\hat{e}$ are more relevant.

DR learners and R-type learners construct pseudo outcomes and then regress them on $\mathbf{x}$ using a flexible CATE model (e.g. Kennedy, 2020; Nie and Wager, 2021). In these methods, causalKANs are naturally used as the final CATE regressor, that is

$$\hat{\tau}(\mathbf{x}) = g_{\text{KAN}}^{(\tau)}(\mathbf{x}), \qquad (31)$$

trained with an orthogonalized loss. After pruning and symbolification, the resulting $\hat{\tau}(\mathbf{x})$ is a closed form expression for the heterogeneous effect that inherits the robustness properties of the underlying DR or R-type learner. If desired, causalKANs can also parameterize nuisance components such as $\hat{e}$ or $\hat{\mu}_t$, but this is optional and should be guided by which functions the practitioner wishes to interpret.

**Summary.** These examples illustrate a general pattern: causalKANs can replace the neural building blocks of many estimators (IPW, meta learners, DR and R learners, etc.) while leaving the estimand (ATE or CATE) unchanged. After pruning and symbolification, the replaced components become analytic, auditable functions of $\mathbf{x}$. The most relevant use of causalKANs is estimator dependent (propensity vs potential outcomes vs CATE) and also dataset dependent, since different applications require understanding different parts of the causal pipeline.

## C  DETAILED RESULTS

### C.1  DETAILS OF THE DATASETS

We have conducted a systematic process of experiments on several datasets. We include here some details of the datasets, and their procedence, for completeness.

**IHDP.**  (Hill, 2011) The Infant Health and Development Program (IHDP) is a semi-synthetic benchmark derived from a real observational study on the effect of early childhood intervention. The publicly used version was constructed by Hill (2011), who retained 25 pre-treatment covariates and simulated potential outcomes to enable pointwise ground truth for individual treatment effects. Following standard practice, we use the canonical 100 replications that define settings A and B. Setting A yields homogeneous treatment effects via linear outcome surfaces, while setting B introduces nonlinear relationships and heterogeneous effects. The dataset contains 747 units with a binary treatment and substantial treatment imbalance, since all treated units from one site were removed in the construction. Covariates include demographic, prenatal, and birth-related variables. All outcomes in the benchmark are synthetic and do not correspond to clinical endpoints.

**ACIC 2016.**  (Dorie et al., 2019) The ACIC 2016 causal inference challenge provides a large collection of semi-synthetic datasets derived from real covariates in the Louisiana Medicaid program. Each instance consists of approximately $N \approx 4800$ units and 58 covariates, including demographic, socioeconomic, and healthcare-related variables. Treatments and potential outcomes are generated through complex nonlinear mechanisms with covariate-dependent assignment, enabling rigorous benchmarking of estimation procedures under realistic confounding. We focus on the nonlinear polynomial and exponential outcome regimes (for example, settings 2, 7, and 26), which are commonly used in prior work. For each selected setting, we use the 77 replications released as part of the competition, each providing a distinct draw of treatment assignment and potential outcomes while keeping covariates fixed across replications. As in other semi-synthetic benchmarks, the outcomes are simulated and have no policy or medical interpretation.

**NSLM.**  The dataset contains covariables from the national study of learning mindset (Yeager et al., 2019). We used the DGP followed by Carvalho et al. (2019), where there are 10000 datapoints (10000 students across 76 schools), with 3 categorical variables and 6 continuous variables, and the treatment and the outcome are simulated with no meaning.

**TCGA.**  The Cancer Genomic Atlas (TCGA) (Weinstein et al., 2013). The TCGA project collected gene expression data fromvarious types of cancers in 9659 individuals. In this case, we conducted the data generating process proposed by Zhang et al. (2023), which keeps only the data from 100 covariates of the RNA sequence. The outcome is simulated and does not have physical meaning. Although in the paper of Zhang et al. (2023), they explore several treatments and dosage, we restrict only a binary treatment without dosage, to have a fair comparison between all the models evaluated.

**NEWS.**  The NEWS dataset (Johansson et al., 2016) was created to perform causal inference. The covariates come from a text database, of 5000 documents, and they represent the count of each word. In total, there are 3477 possible words in each datapoint. The treatment is the device in which the users read, and the outcome is simluated and represent the reader experience. However, we followed the DGP from Crabbé et al. (2022), that employs 100 components of the principal component analysis of the covariates to generate the potential outcomes.

### C.2 ABLATION STUDIES

**Complexity.** We further analyze the relationship between model complexity and the precision in estimation of heterogeneous effects (PEHE). To this end, we define a *complexity score* that aggregates the contributions of different architectural and regularization choices. Specifically, hidden dimensions contribute 0 if no hidden layers are used, 2 if a single hidden layer of size 5 is used, and 3 otherwise. The weight penalty $\lambda$ contributes 0 if set to 0.01 and 1 if set to 0.001, since more regularization yields simpler models. The spline grid contributes 1, 2, or 3 for grid sizes $\{1, 3, 5\}$, respectively. Similarly, the polynomial order $k$ contributes 1, 2, or 3 for $k \in \{1, 3, 5\}$, and sparse initialization reduces the score by one unit. Hence, the complexity score captures the combined effect of the number of hidden units, regularization strength, spline resolution, polynomial order, and initialization strategy. In addition, we compute the number of hidden layers directly from the hidden dimensions: 0 layers (no hidden units, corresponding to KAAM), 1 layer (single hidden layer), and 2 layers (deeper networks).

We then represent the correlation between both complexity and number of hidden layers against PEHE, using linear regression fits and Pearson correlation coefficients across all hyperparameter configurations (Figs. 10 and 11). The results indicate that the correlation is weak: the fitted slopes are close to zero and the Pearson coefficients are generally small. Interestingly, increasing the number of hidden layers often leads to slightly higher PEHE values, although the effect is minor and not consistent across all datasets. Similarly, increasing the overall complexity score does not systematically reduce PEHE.

These findings suggest that, in general, simple KAN architectures achieve competitive PEHE performance, and that increasing architectural or regularization complexity does not yield clear improvements. However, note that this score is a purely descriptive simple heuristic designed to summarize hyperparameter choices and also prioritize more interpretable models.

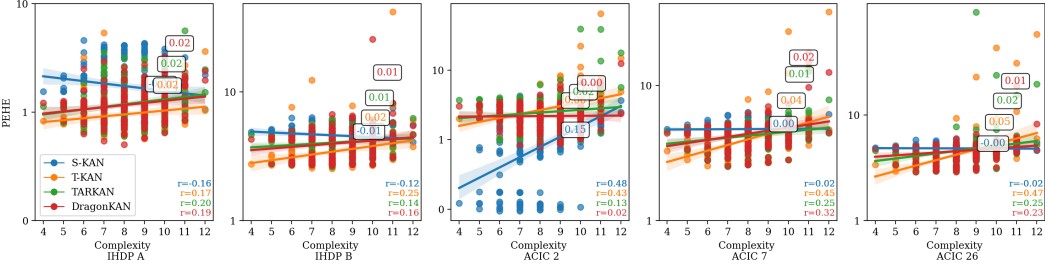

Figure 10: Correlation between model complexity and PEHE across datasets. Each point corresponds to a trained model with a given complexity score. Regression fits (with slopes annotated) and Pearson correlation coefficients are shown. The weak correlations indicate that increasing complexity does not improve PEHE.

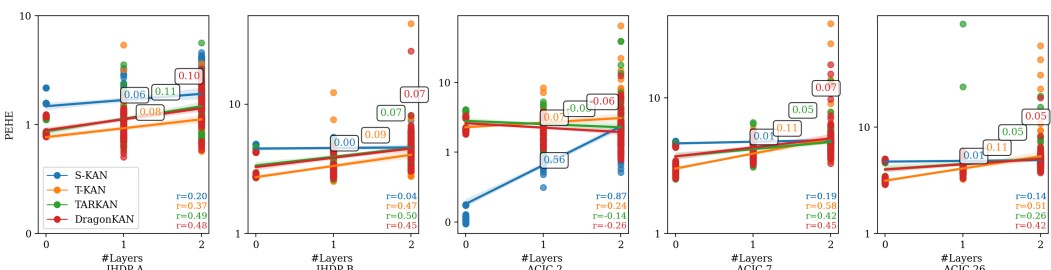

Figure 11: Correlation between the number of hidden layers and PEHE across datasets. The number of layers is computed from the hidden dimensions: 0 (no hidden layers, i.e., KAAM), 1 (single hidden layer), and 2 (two hidden layers). Regression fits and Pearson correlations show that deeper models tend to slightly increase PEHE, although the effect is minor.

**Hyperparameter selection with Loss.** The hyperparameter selection is made based on the validation loss. We want to show that the predictive test loss (i.e. the loss of the KAN model without including sparsity regularizations) is a good proxy for selecting hyperparameters, since the PEHE cannot be computed during evaluation because the real ITE is not known in real-world data and can only be computed with (semi)synthetic data. We can observe in Fig. 12 that these quantities are linearly correlated. However, the study of other surrogate metrics for model selection is still a topic for future work.

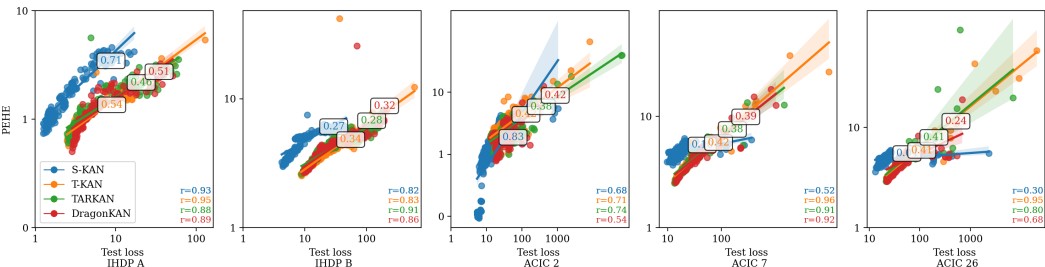

Figure 12: Regression plots of PEHE and Test predictive loss (logarithmic scale) for each dataset with all causalKANs. Slope (in boxes), Pearson coefficient, $r$, (in bottom right corners) and 95% CI (translucent bands) reported. We can observe a clear linear correlation between PEHE and Test loss for all models and all datasets.

Despite this correlation, we acknowledge that outcome-prediction loss is not, in general, a perfect proxy for CATE accuracy, but all models evaluated in our work are potential-outcome regressors trained solely on factual losses. For this family of estimators, the standard practice is to select hyperparameters using validation prediction loss, since these methods do not expose orthogonalized or doubly robust objectives (Shalit et al., 2017; Shi et al., 2019). To ensure a fair comparison, causalKANs use the same training and selection rules.

In Fig. 12, the plateau reflects intrinsic properties of outcome-regression methods, not an issue introduced by causalKANs. Still, we observe strong correlations (typically r > 0.8) between factual test loss and PEHE, indicating that loss remains a model-selection proxy for this class of learners. The slight flattening for the models is consistent with known behavior of non-orthogonalized estimators (Curth and Van der Schaar, 2021b; Shalit et al., 2017), where outcome prediction can improve faster than CATE error.

## C.3 INTERPRETABILITY VISUALIZATIONS

First of all, let us explain in detail the different visualization tools, in addition to the formula analysis *per se*, that will help us to understand outcome variations, depending of the model used.

**Probability Radar Plots (PRPs).** Probability Radar Plots (PRPs) (Saary, 2008) provide an interpretable visualization of generalized additive models (GAMs) by mapping the isolated contribution of each covariate into a radial plot. In our setting, each component function depends on a single covariate, which enables a decomposition of the prediction into additive terms. We denote this decomposition by $\mathbf{\Delta}$.

$$\mathbf{\Delta} = \begin{bmatrix} f_{\mathbf{x}_{1,1}} & f_{\mathbf{x}_{1,2}} & \cdots & f_{\mathbf{x}_{1,D}} \\ f_{\mathbf{x}_{2,1}} & f_{\mathbf{x}_{2,2}} & \cdots & f_{\mathbf{x}_{2,D}} \\ \vdots & \vdots & \ddots & \vdots \\ f_{\mathbf{x}_{N,1}} & f_{\mathbf{x}_{N,2}} & \cdots & f_{\mathbf{x}_{N,D}} \end{bmatrix}, \tag{32}$$

where $f_{\mathbf{x}_{i,j}}$ denotes the contribution of the feature $j$ for individual $i$.

To construct a PRP, we first compute the average contribution of each covariate across all individuals, providing a baseline that reflects the average outcome (for S-KAAM) or the average conditional

average treatment effect (for T-KAAM). Then, for a given individual $i$, we plot the vector of deviations

$$\left(f_{x_{i,1}} - \tfrac{1}{N}\sum_{\ell=1}^{N} f_{x_{\ell,1}}, \, \ldots, \, f_{x_{i,D}} - \tfrac{1}{N}\sum_{\ell=1}^{N} f_{x_{\ell,D}}\right), \tag{33}$$

which highlights how the contribution of each covariate for individual $i$ differs from the population average. By arranging these deviations radially, PRPs enable intuitive comparison across covariates and between individuals.

**Partial dependence plots (PDPs).** We employ partial dependence plots (PDPs) (Friedman, 2001) as visualization tools for interpreting causalKANs. Unlike their conventional use in a "black-box" fashion—where the model is queried without access to its internals—KAN-based PDPs (Almodóvar et al., 2025) directly exploit the analytic equations of the learned model, displaying the underlying splines that constitute the predictors. This provides more faithful and transparent representations of the learned dependencies.

As noted by Loftus et al. (2024), PDPs that vary one covariate in isolation may be misleading in settings with mediators, since they ignore induced changes in other covariates. In our case, however, the assumptions in §3 guarantee that the covariates form a valid adjustment set, excluding mediators. Under these conditions, PDPs are valid tools to represent causal dependencies (Zhao and Hastie, 2021), and in fact rely on the same backdoor adjustment formula (Pearl, 2009) when the effect is homogeneous.

Formally, PDPs display the average variation in the predicted outcome as one covariate is varied, marginalizing over the remaining covariates. Their individual-level counterpart, individual conditional expectation (ICE) curves (Goldstein et al., 2015), provide counterfactual explanations for specific samples. Under additivity, PDPs and ICE coincide; under non-additivity, PDPs correspond to averages of the heterogeneous ICE curves.

### C.3.1 HETEOGENEOUS CATE WITH T-KAAM

We show an example of how T-KAAM captures the CATE as a closed formula that can be interpreted. For the dataset *ACIC 7*, the T-KAAM model is one of the best-performers, and we can observe that the CATE can be represented as an addition of functions of each variable independently (see Eq. 24). An example obtained with one realization of the dataset can be observed in the following equation.

$$
\begin{aligned}
\hat{\tau}(\mathbf{x}) = \ & 0.09\,x_1^2 + 0.19\,x_1 - 0.74\,x_{10}^2 + 5.30\,x_{10} + 0.01\,x_{12}^3 - 0.05\,x_{12}^2 - 0.09\,x_{12} \\
& + 0.03\,x_{13}^3 - 0.13\,x_{13}^2 - 0.21\,x_{13} - 0.06\,x_{14}^2 + 0.33\,x_{14} - 0.06\,x_{15}^2 + 0.13\,x_{15} \\
& + 0.19\,x_{16}^2 - 0.22\,x_{16} - 0.59\,x_{17} - 0.03\,x_{18}^3 - 0.14\,x_{18}^2 - 0.47\,x_{18} \\
& + 0.03\,x_{19}^4 - 0.10\,x_{19}^3 - 0.33\,x_{19}^2 + 0.42\,x_{19} + 0.14\,x_{20}^3 - 0.29\,x_{20}^2 - 0.30\,x_{20} \\
& + 0.01\,x_{21}^3 + 0.05\,x_{21}^2 + 0.28\,x_{21} + 0.25\,x_{23}^3 - 0.50\,x_{23}^2 - 0.83\,x_{23} - 0.16\,x_{24} \\
& - 0.07\,x_{25}^2 + 0.14\,x_{25} - 1.42\,x_{26} - 0.01\,x_{27}^4 + 0.04\,x_{27}^3 - 0.06\,x_{27}^2 + 0.06\,x_{27} \\
& - 0.12\,x_{28}^2 - 0.32\,x_{28} + 0.00\,x_{29}^4 + 0.08\,x_{29}^3 + 0.07\,x_{29}^2 + 0.01\,x_{29} \\
& - 0.16\,x_3 - 0.17\,x_{30} - 0.01\,x_{31}^2 - 0.08\,x_{31} + 0.54\,x_{33} \\
& - 0.03\,x_{34}^2 - 0.17\,x_{34} + 0.02\,x_{35}^3 + 0.23\,x_{35}^2 + 0.84\,x_{35} \\
& + 0.03\,x_{36}^3 - 0.14\,x_{36}^2 + 0.22\,x_{36} - 0.30\,x_{38} - 0.15\,x_{39}^2 - 0.20\,x_{39} \\
& - 0.03\,x_4^2 + 0.12\,x_4 - 0.13\,x_{40}^3 - 0.31\,x_{40}^2 - 0.18\,x_{40} \\
& + 0.04\,x_{41}^3 - 0.39\,x_{41}^2 + 0.02\,x_{41} + 0.25\,x_{42}^2 + 0.18\,x_{42} \\
& + 0.01\,x_{43}^4 - 0.03\,x_{43}^3 - 0.09\,x_{43}^2 + 0.08\,x_{43} \\
& - 0.41\,x_{44}^3 - 2.42\,x_{44}^2 - 0.92\,x_{44} - 0.06\,x_{45}^3 - 0.29\,x_{45}^2 - 0.32\,x_{45} \\
& - 0.09\,x_{46}^2 - 0.04\,x_{46} - 1.10\,x_{49} - 0.10\,x_5^4 + 0.56\,x_5^3 + 1.27\,x_5^2 - 1.30\,x_5 \\
& - 0.02\,x_{50}^2 + 0.01\,x_{50} - 0.12\,x_{51} + 0.14\,x_{52}^2 - 0.71\,x_{52} - 0.57\,x_{53} + 0.42\,x_{54} \\
& - 0.02\,x_{56}^3 + 0.01\,x_{56}^2 + 0.52\,x_{56} - 0.02\,x_{57}^3 + 0.09\,x_{57}^2 + 0.23\,x_{57} \\
& - 0.02\,x_{58}^3 - 0.17\,x_{58}^2 - 0.34\,x_{58} - 0.06\,x_7^2 + 0.41\,x_7 - 0.21\,x_8 + 0.21\,x_9^2 - 0.25\,x_9 \\
& + 9.47\,.
\end{aligned}
$$

This formula is long, due to the high number of covariates. Therefore, although the contribution of each variable has a polynomial form, analyzing the formula alone could be tricky. To help to gain intuitions about the contribution of each variable to the CATE variation, we present three useful plots in Fig. 13. In the plots, we can obtain the contributions of each feature to the causal effect. However, those visualizations should not be used to intervene in any feature except the treatment, since the variations presented in Fig. 13 cannot be interpreted as causal effect following our assumptions.

Lastly, we want to show that our *inductive bias* for simplicity of the atom selection is useful to improve interpretability. The formula below is provided by the standard auto-symbolic function (native of pyKAN (Liu et al., 2024b)). We observe polynomial relations with some covariates, but the nonlinear complex functions that follow are less interpretable than simple polynomials.

$$
\begin{aligned}
\hat{\tau}(\mathbf{x}) = \ & -0.59\,x_{17} - 0.16\,x_{20} + 0.26\,x_{21} - 1.41\,x_{26} + 0.10\,x_{27} + 0.28\,x_{29} \\
& - 0.25\,x_{31} + 0.54\,x_{33} + 0.15\,x_{34} + 0.34\,x_{35} + 0.22\,x_{36} - 0.30\,x_{38} \\
& - 0.32\,x_{40} + 0.02\,x_{43} - 1.67\,x_{44} - 0.40\,x_{45} + 0.65\,x_5 - 0.12\,x_{51} \\
& - 0.57\,x_{53} + 0.42\,x_{54} - 0.02\,x_{55} - 0.40\,x_{58} \\
& - 0.00\,(1.17 - 9.05\,x_{50})^2 + 0.01\,(3.95 - 3.98\,x_{16})^2 \\
& - 0.00\,(5.62 - 7.80\,x_{41})^2 - 0.19\,(6.55 - 1.78\,x_{10})^2 \\
& - 0.02\,(7.66 - 2.42\,x_{10})^2 + 0.01\,(9.47 - 3.82\,x_{52})^2 \\
& + 0.00\,(9.80 - 4.97\,x_{46})^2 + 0.00\,(-2.05\,x_{16} - 5.95)^2 \\
& - 0.00\,(-9.30\,x_{23} - 2.38)^2 + 0.12\,\sqrt{5.29\,x_{31} + 2.54} \\
& - 0.00\,(-8.38\,x_{39} - 5.79)^2 - 1.03\,\exp(0.60\,x_{18}) - 0.23\,\exp(0.97\,x_{24}) \\
& - 0.53\,\exp(0.56\,x_{34}) + 0.47\,\exp(0.87\,x_{35}) + 1.12\,\sin(0.63\,x_1 - 7.20) \\
& - 0.40\,\sin(0.81\,x_1 - 0.81) - 0.24\,\sin(4.88\,x_{12} + 1.30) \\
& - 0.40\,\sin(0.83\,x_{13} - 1.02) + 1.09\,\sin(5.19\,x_{13} + 5.19) \\
& - 0.33\,\sin(1.81\,x_{14} + 2.36) + 2.42\,\sin(9.01\,x_{15} + 7.79) \\
& - 0.56\,\sin(1.35\,x_{19} - 2.19) - 0.26\,\sin(1.14\,x_{20} + 1.39)
\end{aligned}
$$

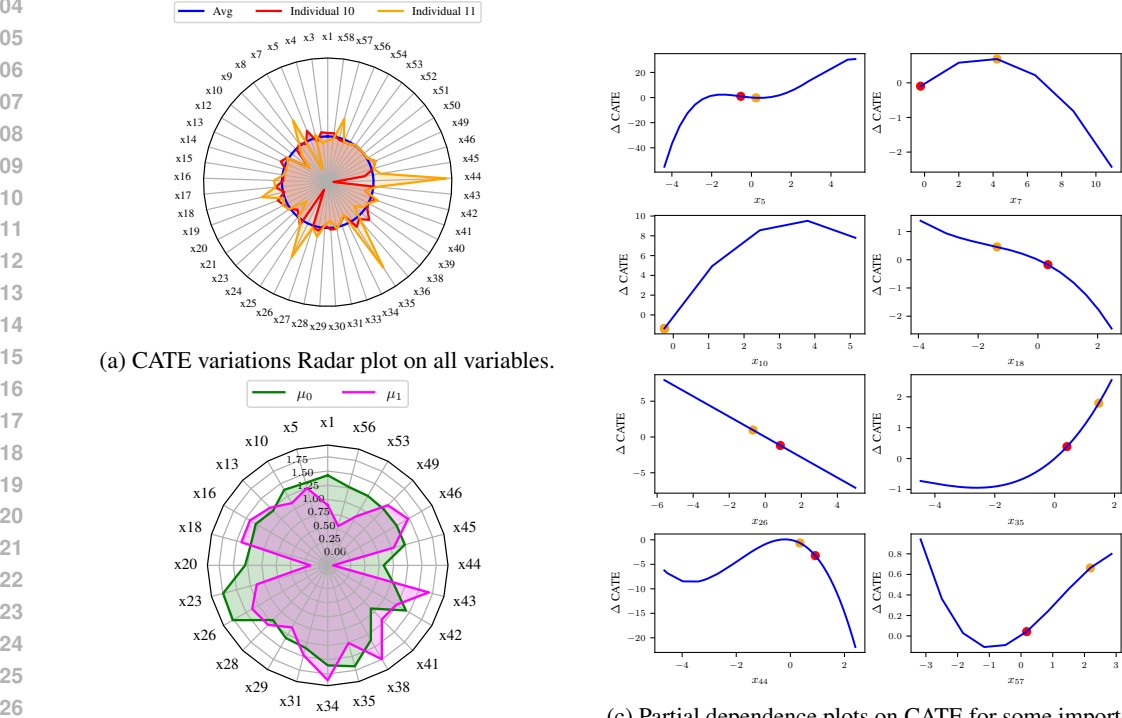

(a) CATE variations Radar plot on all variables.

(b) Potential outcome Radar plot for Individual 11.

(c) Partial dependence plots on CATE for some important variables.

Figure 13: Visualization plots employing T-KAAM with ACIC-7. **(a)** *Radar plot* of the contribution of each variable to the CATE. In blue, the average predicted CATE in all the dataset. In red and orange, the respective contributions of each variable of individuals 10 and 11. **(b)** *Radar plot* of the potential outcomes of the individual 10. We can observe the contribution of each variable to the potential outcomes. Only the variables that have not been pruned in both subnetworks have been added to the plot. **(c)** *Partial dependence plots* of the CATE variation, given variations on the most important variables in the PRP. Particular values of CATE variation for the individuals 10 and 11 shows their contribution.

$$+ 0.51 \sin(1.80 \, x_{23} - 10.00) - 0.50 \sin(0.63 \, x_{25} - 8.42)$$
$$- 0.86 \sin(0.48 \, x_{28} - 4.36) + 2.25 \sin(0.52 \, x_{28} + 1.99)$$
$$+ 1.09 \sin(0.22 \, x_3 + 8.43) - 1.02 \sin(4.20 \, x_{30} + 1.41)$$
$$+ 0.58 \sin(7.77 \, x_{41} - 3.80) + 0.74 \sin(1.04 \, x_{42} - 8.42)$$
$$+ 0.75 \sin(5.35 \, x_{42} + 0.85) + 1.29 \sin(9.59 \, x_{46} + 3.61)$$
$$+ 0.53 \sin(4.58 \, x_{49} + 2.43) - 4.78 \sin(4.79 \, x_{49} - 0.60)$$
$$+ 0.88 \sin(4.21 \, x_{56} + 7.78) - 1.94 \sin(5.59 \, x_{56} - 7.58)$$
$$+ 0.41 \sin(0.88 \, x_{57} - 7.16) + 1.03 \sin(3.20 \, x_7 + 7.20)$$
$$+ 0.72 \sin(0.40 \, x_8 + 2.00) + 1.07 \sin(9.06 \, x_9 + 1.76)$$
$$+ 0.53 \tanh(0.98 \, x_{18} - 1.40) + 7.14 - 0.11 \exp(-0.93 \, x_4) \, .$$

### C.3.2 HOMOGENEOUS CATE WITH S-KAAM

In this section, we illustrate how we obtain the *homogeneous* CATE (or, equivalently, the ATE) in the IHDP A dataset, where the causal effect of the treatment is known to be homogeneous and linear.

Therefore, we instantiate an S-KAAM, which yields the following formula in the potential outcome estimation.

$$\hat{\mu}(\mathbf{x}, \mathbf{t}) = \underline{3.74 \, \mathbf{t}} + 0.53 \, x_1 + 0.16 \, x_{10} + 0.59 \, x_{11} - 0.11 \, x_{12} + 0.34 \, x_{13} + 0.11 \, x_{16} - 0.17 \, x_{18} + 1.28 \, x_{19}$$
$$+ 0.01 \, x_2^3 - 0.01 \, x_2^2 - 0.23 \, x_{20} - 0.08 \, x_{21} + 0.11 \, x_{24} + 0.18 \, x_3^2 + 1.29 \, x_3 + 0.28 \, x_5$$

$$+ 0.01\,x_6^4 - 0.03\,x_6^3 + 0.03\,x_6 + 1.47\,x_8 + 0.32\,x_9 + 1.88\,.$$

As presented in Eq. 23, the CATE can be computed exclusively with the terms relative to **t**. In this case, the CATE can be directly extracted from the formula: 3.74.

We also represent in Fig. 14 some visualizations that we find interesting. First, for two given individuals, we present a PRP with the contribution of each variable to the variation of the predicted outcome, $\hat{\mu}(\mathbf{x}, \mathbf{t})$, compared with the average of the predicted outcomes, $\mathbb{E}_{\mathbf{x}, \mathbf{t}}[\hat{\mu}(\mathbf{x}, \mathbf{t})]$. On the right, we present the predicted potential outcomes for a given individual. As the effect is homogeneous (does not depend on the covariates), the contribution of each feature for both potential outcomes is the same, and the only difference is the causal effect of the treatment. Lastly, we present PDPs for treatment and other three variables (based on the radar plot), which present the variations of the predicted outcome with each variable. Following our assumptions, only the treatment curve can be seen as a causal effect, while the other curves can be used only to gain intuition of outcome behavior.

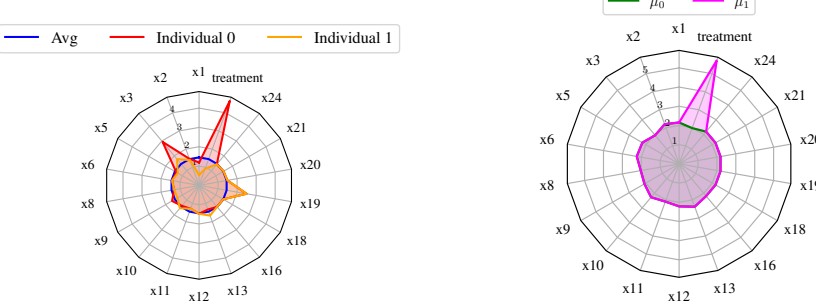

(a) Outcome variations Radar plot on all variables. (b) Potential outcome Radar plot for Individual 0.

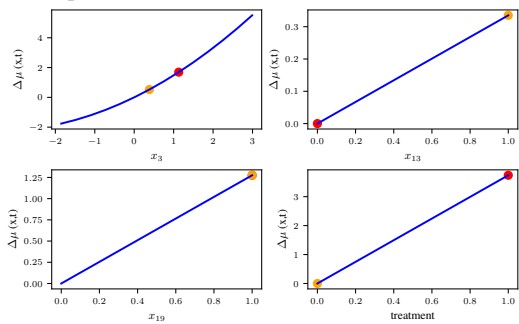

(c) Partial dependence plots on CATE for some important variables.

Figure 14: Visualization plots employing S-KAAM with IHDP A. **(a)** *Radar plot* of the contribution of each variable to the outcome. In blue, the average predicted outcome in all the dataset. In red and orange, the respective contributions of each variable of individuals 0 and 1. **(b)** *Radar plot* of the potential outcomes of the individual 0. We can observe the contribution of each variable to potential outcomes. **(c)** *Partial dependence plots* of the outcome variation, given variations on the most important variables in the PRP. Particularized for the individuals 0 and 1.

In this case, the standard symbolic substitution provides a similar formula as our proposal, due to the linearity of the dataset, and we omit its expression.

### C.3.3 METRIC VARIATIONS FOLLOWING THE PIPELINE

We also show how the metrics—both observed (test loss) and unobserved (PEHE) in real data—vary when we perform the pruning and the symbolic substitution in the examples that we develop in App. C.3.

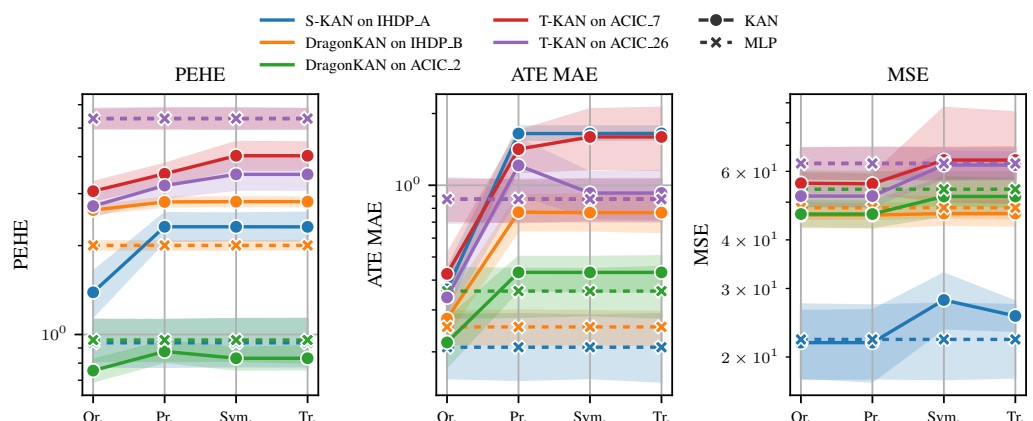

Figure 15: Variation of the metrics in each step of the pipeline: original (Or.), pruned (Pr.), Formula (For.) and 2-decimal truncation (Tr.).

We can observed in Fig. 15, for each dataset (with its respective model), the variation in performance when **i)** we prune the network, **ii)** we substitute the splines by symbolic formulas and **iii)** we truncate the formulas to have 2 decimals.

The conclusion of this experiment is that the metric variation in the different steps is high, so a practitioner should be careful when setting the budgets of step acceptance. MSE still show signs of being a good proxy of the PEHE, representing the variations of the PEHE relatively well.

Note that, when following the pipeline, large errors can occur (e.g. when pruning S-KAN). A practitioner would either adjust the pruning threshold to limit changes in the estimated ITEs or simply reject pruning when it induces excessive error. For these experiments, we used the default pruning and symbolic substitution criteria ($\Gamma = 3 \cdot 10^{-2}$, $\Gamma_{R^2} = 0.98$) because the goal is to evaluate pruning uniformly across all benchmark datasets; tuning it per-dataset would create an unfair comparison.

### C.4 EXPERIMENTS OF EXPRESSION RECOVERY

In this section, we add details about the experiments of CATE recovery, explained in §5.3. There, we have shown that for some expressions that can be captured by S-KAAM and T-KAAM respectively, the pipeline of causalKANs achieves better approximations of the true data-generating functions, than MLP or KANs without simplification steps. Those steps act as inductive biases towards a simpler representation.

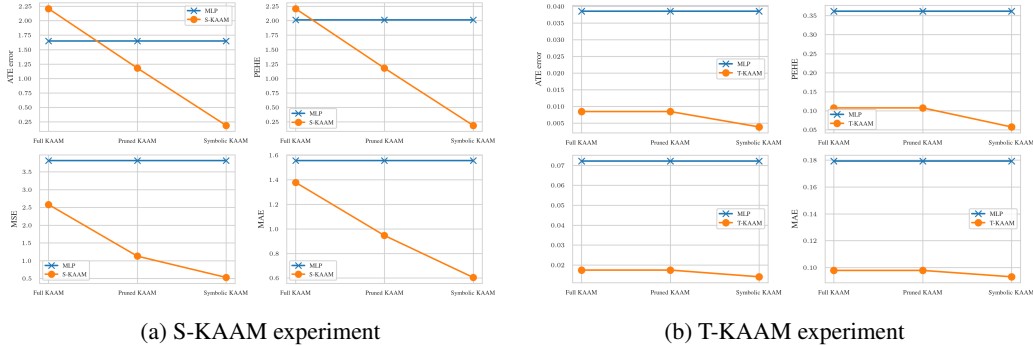

(a) S-KAAM experiment        (b) T-KAAM experiment

Figure 16: Metrics of the synthetic experiments that show function recovery: ATE error, PEHE, MSE and MAE. ATE error and PEHE can only be computed observing the counterfactual, while MAE and MSE are functions of observational data. **Lower is better.**

We can observe in Fig. 16 that all metrics decrease (improve) while computing the simplification steps, specially the symbolic substitution. That information is complementary to the curves observed in Fig. 6, that show a better fit of the symbolic curves. After symbolic substitution, the PEHE and the ATE error are much lower than the achieved by the causalNN counterpart in each case.

### C.5 TIME CONSUMPTION

We are interested in comparing training and inference time of causalKANs with their respective causalNNs. It is well known that KANs take more time to train than MLPs, for networks with the same number of parameters. However, KANs usually require less parameters than MLPs to achieve similar performance (Liu et al., 2024b). For the tasks analyzed in this paper, causalKANs requires greater computational effort during both training and inference than existing causal neural networks. We consider this additional cost justified by the interpretability that causalKANs provides, which we regard as a central advantage for causal analysis. We report the relative training and inference times in Tab 2.

| Dataset | Model Name | KAN Training (s) | KAN Inference (s) | MLP Training (s) | MLP Inference (s) | Ratio (Train) KAN/MLP |
|---|---|---|---|---|---|---|
| IHDP A | S-Learner | $23.62_{19.25}$ | $0.12_{0.02}$ | $1.98_{2.05}$ | $0.00_{0.00}$ | 12 |
| | T-Learner | $97.89_{75.57}$ | $0.03_{0.01}$ | $29.00_{29.19}$ | $0.00_{0.00}$ | 7/2 |
| | TarNet | $67.21_{52.39}$ | $0.01_{0.00}$ | $8.02_{9.27}$ | $0.00_{0.00}$ | 8 |
| | DragonNet | $62.92_{48.39}$ | $0.01_{0.00}$ | $9.74_{10.28}$ | $0.00_{0.00}$ | 6/1 |
| IHDP B | S-Learner | $56.24_{22.13}$ | $0.06_{0.01}$ | $2.61_{1.24}$ | $0.00_{0.00}$ | 22 |
| | T-Learner | $58.49_{14.10}$ | $0.03_{0.00}$ | $45.81_{18.08}$ | $0.00_{0.00}$ | 9/7 |
| | TarNet | $31.39_{10.88}$ | $0.03_{0.00}$ | $42.37_{16.43}$ | $0.00_{0.00}$ | 3/4 |
| | DragonNet | $157.82_{38.35}$ | $0.01_{0.00}$ | $187.26_{31.89}$ | $0.00_{0.00}$ | 5/6 |
| ACIC 2 | S-Learner | $120.96_{54.54}$ | $0.05_{0.01}$ | $23.13_{15.77}$ | $0.00_{0.00}$ | 5 |
| | T-Learner | $368.94_{153.74}$ | $0.22_{0.03}$ | $162.59_{28.60}$ | $0.01_{0.00}$ | 9/4 |
| | TarNet | $135.38_{83.87}$ | $0.05_{0.01}$ | $55.08_{62.84}$ | $0.01_{0.00}$ | 5/2 |
| | DragonNet | $92.83_{55.77}$ | $0.04_{0.01}$ | $63.17_{70.06}$ | $0.01_{0.00}$ | 3/2 |
| ACIC 7 | S-Learner | $147.31_{110.57}$ | $0.23_{0.03}$ | $4.56_{7.80}$ | $0.00_{0.00}$ | 32 |
| | T-Learner | $377.66_{177.18}$ | $0.04_{0.01}$ | $39.90_{43.24}$ | $0.01_{0.00}$ | 9 |
| | TarNet | $236.88_{109.75}$ | $0.03_{0.00}$ | $40.38_{50.22}$ | $0.01_{0.00}$ | 6 |
| | DragonNet | $272.19_{127.14}$ | $0.03_{0.00}$ | $51.86_{65.79}$ | $0.01_{0.00}$ | 5 |
| ACIC 26 | S-Learner | $141.11_{96.81}$ | $0.17_{0.02}$ | $4.46_{7.54}$ | $0.00_{0.00}$ | 32 |
| | T-Learner | $145.50_{70.85}$ | $0.03_{0.00}$ | $39.36_{42.07}$ | $0.01_{0.00}$ | 7/2 |
| | TarNet | $111.13_{53.19}$ | $0.02_{0.00}$ | $39.89_{49.73}$ | $0.01_{0.00}$ | 8/3 |
| | DragonNet | $147.49_{70.45}$ | $0.03_{0.05}$ | $48.26_{58.27}$ | $0.01_{0.00}$ | 3 |
| NSLM | S-Learner | $75.62_{44.37}$ | $0.38_{0.09}$ | $73.57_{29.65}$ | $0.01_{0.00}$ | 1 |
| | T-Learner | $12.14_{3.92}$ | $0.06_{0.01}$ | $155.73_{89.40}$ | $0.01_{0.00}$ | 1/13 |
| | TarNet | $10.70_{2.46}$ | $0.07_{0.01}$ | $130.49_{63.07}$ | $0.00_{0.00}$ | 1/12 |
| | DragonNet | $19.61_{3.92}$ | $0.06_{0.01}$ | $214.24_{34.63}$ | $0.00_{0.00}$ | 1/11 |
| NEWS | S-Learner | $40.28_{18.32}$ | $0.24_{0.04}$ | $2.53_{0.43}$ | $0.01_{0.00}$ | 16 |
| | T-Learner | $227.39_{94.20}$ | $0.23_{0.28}$ | $147.36_{77.26}$ | $0.00_{0.00}$ | 3/2 |
| | TarNet | $45.75_{17.19}$ | $0.08_{0.17}$ | $5.87_{0.50}$ | $0.00_{0.00}$ | 8 |
| | DragonNet | $19.32_{5.97}$ | $0.06_{0.01}$ | $6.54_{0.41}$ | $0.00_{0.00}$ | 3 |
| TCGA | S-Learner | $122.09_{37.81}$ | $0.31_{0.23}$ | $48.05_{18.54}$ | $0.01_{0.00}$ | 5/2 |
| | T-Learner | $269.33_{101.08}$ | $0.19_{0.12}$ | $778.61_{338.08}$ | $0.00_{0.01}$ | 1/3 |
| | TarNet | $285.42_{117.24}$ | $0.10_{0.06}$ | $246.19_{109.33}$ | $0.01_{0.00}$ | 6/5 |
| | DragonNet | $185.98_{100.10}$ | $0.09_{0.02}$ | $31.35_{2.40}$ | $0.01_{0.00}$ | 6 |

Table 2: Comparison of training and inference times (in seconds) for KAN and MLP across datasets. The last column shows the ratio of KAN to MLP training time, approximated as simple fractions. Values are reported as mean$_{std}$.

In average, causalKANs train slower than causalNNs: KAN training time $\approx 8 \times$ MLP training time. However, note that the time consumption depends largely on the dataset, since we can observe that, for the NSLM dataset, causalKANs trains much faster than MLP.

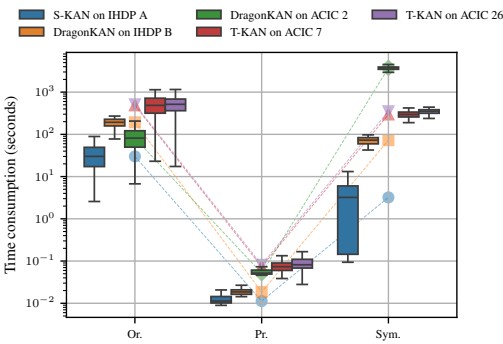

Figure 17: Training of original network (Or.), pruning (Pr.) and symbolic (Sym.) time consumption in seconds for IHDP and ACIC datasets. Logarithmic scale. Boxplots represent the distribution of times in all realizations. Markers and lines represent the medians of each distribution.

**Pipeline time consumption.** Besides training, the rest of the interpretability pipeline (Pruning and symbolic substitution) also introduces complexity in terms of computation. We include in Fig. 17 the distribution of training (Or.), pruning (Pr.) and symbolic substitution (Sym.) times for the best causalKANs in the ACIC and IHDP datasets. All these training times have been measured in a CPU AMD Ryzen Threadripper 7970X 32-Cores. We can observe that the symbolic substitution time is comparable or even higher than training time in some datasets. On the other hand, pruning time is negligible.

## C.6 COMPARISON VISUALIZATIONS

In the same fashion, we have compared each causalNN with its respective causalKAN , in Fig. 18.

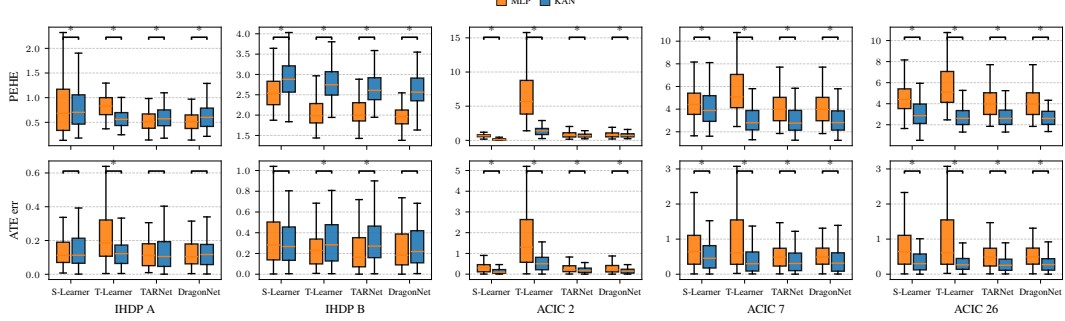

Figure 18: Comparison of KAN vs MLP across datasets. Top row: PEHE, bottom row: ATE err. The trend of the difference is not constant across datasets. For example, we can observe that causalKANs achieve better PEHE metrics in IHDP A and ACIC 2/7/26, but worse in IHDP B, than their respective causalNNs.

We also include, in Tab 3, p-values for transparency, as they provide a more nuanced understanding of the evidence against the null hypothesis than a binary significant/non-significant determination at a fixed $\alpha$ level.

## D COMPLETE PIPELINE

We offer two different complete visualizations of the pipeline that we propose, in an algorithm Alg. 1 version that details all the steps, including the details and the computational step of each block, and a block diagram Fig. 19 in which is easier to focus on the decision steps and the importance of the budgets.

| Dataset | Architecture | KAN | | MLP | |
|---|---|---|---|---|---|
| | | $p$ (ATE err) | $p$ (PEHE) | $p$ (ATE err) | $p$ (PEHE) |
| IHDP A | S-Learner | **0.781** | **0.121** | **0.132** | $< 10^{-3}$ |
| | T-Learner | **0.889** | **1.000** | $< 10^{-3}$ | $< 10^{-3}$ |
| | TarNet | **1.000** | **1.000** | **0.906** | **1.000** |
| | DragonNet | **0.906** | **1.000** | **1.000** | **1.000** |
| IHDP B | S-Learner | 0.006 | **$< 10^{-3}$** | 0.012 | $< 10^{-3}$ |
| | T-Learner | 0.007 | $< 10^{-3}$ | **0.388** | **0.189** |
| | TarNet | 0.003 | $< 10^{-3}$ | **1.000** | **0.189** |
| | DragonNet | **0.388** | **0.006** | **0.436** | **1.000** |
| ACIC 2 | S-Learner | **1.000** | **1.000** | $< 10^{-3}$ | $< 10^{-3}$ |
| | T-Learner | $< 10^{-3}$ | $< 10^{-3}$ | $< 10^{-3}$ | $< 10^{-3}$ |
| | TarNet | 0.013 | $< 10^{-3}$ | $< 10^{-3}$ | $< 10^{-3}$ |
| | DragonNet | **0.157** | $< 10^{-3}$ | $< 10^{-3}$ | $< 10^{-3}$ |
| ACIC 7 | S-Learner | 0.020 | $< 10^{-3}$ | $< 10^{-3}$ | $< 10^{-3}$ |
| | T-Learner | **0.908** | **0.788** | $< 10^{-3}$ | $< 10^{-3}$ |
| | TarNet | **1.000** | **0.788** | 0.020 | $< 10^{-3}$ |
| | DragonNet | **0.706** | **1.000** | $< 10^{-3}$ | $< 10^{-3}$ |
| ACIC 26 | S-Learner | **0.838** | 0.022 | $< 10^{-3}$ | $< 10^{-3}$ |
| | T-Learner | **1.000** | **0.511** | $< 10^{-3}$ | $< 10^{-3}$ |
| | TarNet | **1.000** | **0.511** | 0.011 | $< 10^{-3}$ |
| | DragonNet | **1.000** | **1.000** | $< 10^{-3}$ | $< 10^{-3}$ |
| NSLM | S-Learner | **1.000** | **1.000** | $< 10^{-3}$ | $< 10^{-3}$ |
| | T-Learner | **0.15** | **1.000** | **0.52** | $< 10^{-3}$ |
| | TarNet | 0.01 | **1.000** | **0.44** | $< 10^{-3}$ |
| | DragonNet | $< 10^{-3}$ | **0.104** | **0.196** | $< 10^{-3}$ |
| NEWS | S-Learner | **0.621** | **0.406** | $< 10^{-3}$ | $< 10^{-3}$ |
| | T-Learner | **0.506** | $< 10^{-3}$ | $< 10^{-3}$ | 0.006 |
| | TarNet | **1.000** | 0.006 | **0.101** | **1.000** |
| | DragonNet | **0.621** | 0.003 | $< 10^{-3}$ | **0.488** |
| TCGA | S-Learner | $< 10^{-3}$ | $< 10^{-3}$ | $< 10^{-3}$ | 0.039 |
| | T-Learner | $< 10^{-3}$ | $< 10^{-3}$ | **1.000** | $< 10^{-3}$ |
| | TarNet | $< 10^{-3}$ | $< 10^{-3}$ | $< 10^{-3}$ | **1.000** |
| | DragonNet | $< 10^{-3}$ | $< 10^{-3}$ | $< 10^{-3}$ | 0.001 |

Table 3: Friedman post-hoc *corrected* $p$-values per dataset and metric, arranged to mirror Tab 1. Baselines are underlined (reported as $p = 1.000$) and methods not significantly different from the baseline at $\alpha = 0.05$ are in **bold**. Very small $p$-values are reported as $< 10^{-3}$.

As can be observed, although the pipeline is well defined, the are many variables (or hyperparameters) that the practitioner should vary depending on the dataset or the application of our proposal We include here some recommendations to choose these parameters and some discussion about the pipeline.

First of all, note that we treat KANs as a modeling option, not a universally superior alternative. Therefore, the practitioner should decide in each step if the causalKAN should be used or if one should keep the causalNN baseline. For example, regarding the benchmarking experiments of §5, we observe that TCGA obtains ITEs that are very far from the causalNNs estimates. Therefore, the practitioner should evaluate if that is acceptable. The same criteria should be applied with the rest of the thresholds. For example, when pruning the DragonKAN in IHDP B in Fig. 15, we observe that both PEHE and ATE move away from causalNN estimates. A practitioner should establish a threshold of admissible deviation. However note that the variation in metrics given by the pipeline steps are dependant of the parameters established the methods applied: in the pruning step, the pruning threshold $\Gamma$; in the symbolic substitution, the $\Gamma_{R^2}$. We recommend to test several of these parameters, and compare the deviation with the pipeline budgets ($\Lambda$). Therefore, the process of selecting those parameters can be cyclic and the practitioner should establish the number of iterations that are admissible. For the experiments, we set the default parameters of pruning (edge threshold

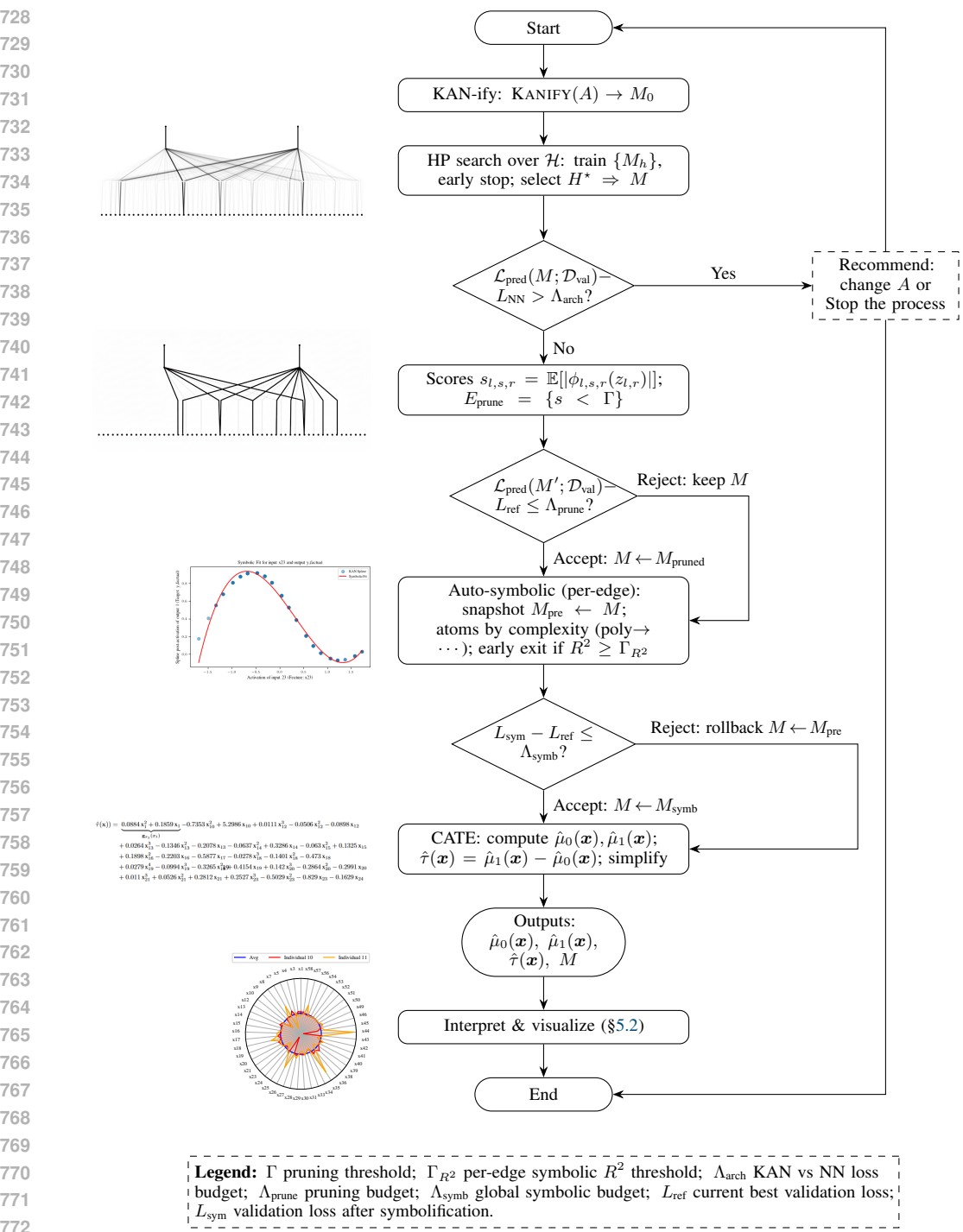

Figure 19: Block diagram of CausalKAN pipeline with explicit Accept/Reject semantics and final outputs.

$3 \cdot 10^{-2}$, node threshold $10^{-2}$) and $R^2$ ($\Gamma_{R^2}$=0.98) for getting a fair comparison. Therefore, the accept-reject gates are not static steps, but should be defined depending on the dataset and task.

## E    LLM USAGE

Large language models (LLMs) were used exclusively to improve the clarity and grammar of the manuscript, and to assist with the presentation of plots and figures in Python and LaTeX (TikZ). All scientific contributions, methodological developments, experiments, and analyses were conceived, implemented, and validated by the authors, with no use of LLMs.

---

**Algorithm 1** CausalKAN : Interpretable CATE via KAN-ified causal networks

---

**Input:** Base causalNN $A$; splits $\mathcal{D}_{\text{tr}}, \mathcal{D}_{\text{val}}, \mathcal{D}_{\text{te}}$ with $(\mathbf{x}, \mathbf{t}, \mathbf{y})$; budgets $\Lambda_{\text{prune}}, \Lambda_{\text{symb}}$; thresholds $\Gamma$ (prune), $\Gamma_{R^2}$ (symbolic), $\Lambda_{\text{arch}}$ (KAN vs NN); HP spaces, $\mathcal{H}$, (depth, width, spline grid, $\lambda_1, \lambda_c, \lambda_s, \lambda_H$); atom dict $\{f_m\}_{m=1}^M$ ordered by complexity (poly $\rightarrow$ trigs $\rightarrow$ others); optimizer, early stopping.

**Output:** $\hat{\mu}_0(\boldsymbol{x}), \hat{\mu}_1(\boldsymbol{x}), \hat{\tau}(\boldsymbol{x}) \equiv \hat{\mu}_1(\boldsymbol{x}) - \hat{\mu}_0(\boldsymbol{x})$

1: **KAN-ification:** $M_0 \leftarrow \textsc{Kanify}(A)$

2: **HP search & training:**
3: **for** $h \in \mathcal{H}$ **do**
4:      $M_h \leftarrow \textsc{Instantiate}(M_0, h)$
5:      Minimize on $\mathcal{D}_{\text{tr}}$:

$$\mathcal{L} = \mathcal{L}_{\text{pred}}(M_h) + \lambda_1 \sum_{l,s,r} \mathbb{E}[|\phi_{l,s,r}(z_{l,r})|] + \lambda_c \sum |w_s| + \lambda_s \sum |w_s - w_s'| + \lambda_H \sum H$$

6:      Early stop on $\mathcal{D}_{\text{val}}$; store $L_{\text{val}}(h) = \mathcal{L}_{\text{pred}}(M_h^\dagger; \mathcal{D}_{\text{val}})$
7: **end for**
8: $H^\star \leftarrow \arg\min_h L_{\text{val}}(h)$ with tie-break by simplicity (fewer layers, smaller grids/nodes, no MultKAN)
9: $M \leftarrow M_{H^\star}^\dagger;\quad L_{\text{ref}} \leftarrow L_{\text{val}}(H^\star)$

10: **Baseline causalNN check (KAN vs NN):**
11: Train $A^\dagger$ (original causalNN) under its best HPs on $\mathcal{D}_{\text{tr}}$ with early stopping
12: $L_{\text{NN}} \leftarrow \mathcal{L}_{\text{pred}}(A^\dagger; \mathcal{D}_{\text{val}})$
13: **if** $\mathcal{L}_{\text{pred}}(M; \mathcal{D}_{\text{val}}) - L_{\text{NN}} > \Lambda_{\text{arch}}$ **then**
14:      **Warn/Recommend:** change $A$ (architecture underperforming);
15: **end if**

16: **Pruning (accept–reject):**
17: $s_{l,s,r} \leftarrow \mathbb{E}_{\mathcal{D}_{\text{val}}}[|\phi_{l,s,r}(z_{l,r})|]$
18: $E_{\text{prune}} \leftarrow \{(l, s \rightarrow r) : s_{l,s,r} < \Gamma\}$
19: **if** $E_{\text{prune}} \neq \emptyset$ **then**
20:      $M' \leftarrow \textsc{RemoveEdgesAndIsolatedNodes}(M, E_{\text{prune}})$
21:      **if** $\mathcal{L}_{\text{pred}}(M'; \mathcal{D}_{\text{val}}) - L_{\text{ref}} \leq \Lambda_{\text{prune}}$ **then**
22:          $M \leftarrow M'; L_{\text{ref}} \leftarrow \mathcal{L}_{\text{pred}}(M; \mathcal{D}_{\text{val}})$
23:      **else**
24:          **Reject** pruning
25:      **end if**
26: **end if**

27: **Auto-symbolic (per-edge, early exit by $R^2$) + global accept:**
28: $M_{\text{pre}} \leftarrow M$                            $\triangleright$ snapshot for possible rollback
29: **for** each edge $e = (l, s \rightarrow r)$ in $M$ **do**
30:      collect $\{(u_k, v_k)\}$ on $\mathcal{D}_{\text{val}}$: $u_k = z_{l,r}^{(k)}$, $v_k = \phi_{l,s,r}(u_k)$
31:      **for** $m = 1 \rightarrow M$ **(complexity-ordered) do**
32:          $(a^\star, b^\star, c^\star, d^\star) \leftarrow \arg\min_{a,b,c,d} \frac{1}{|\mathcal{D}_{\text{val}}|} \sum_k \left(v_k - [cf_m(au_k + b) + d]\right)^2$
33:          compute $R_m^2$ on $\mathcal{D}_{\text{val}}$
34:          **if** $R_m^2 \geq \Gamma_{R^2}$ **then**
35:              $M \leftarrow \textsc{ReplaceEdgeWithAtom}(M, e, f_m, a^\star, b^\star, c^\star, d^\star)$      $\triangleright$ early accept for this edge
36:              **break**
37:          **end if**
38:      **end for**
39: **end for**
40: $L_{\text{sym}} \leftarrow \mathcal{L}_{\text{pred}}(M; \mathcal{D}_{\text{val}})$
41: **if** $L_{\text{sym}} - L_{\text{ref}} \leq \Lambda_{\text{symb}}$ **then**
42:      **Accept** symbolic model; $L_{\text{ref}} \leftarrow L_{\text{sym}}$
43: **else**
44:      **Reject** symbolic model; $M \leftarrow M_{\text{pre}}$
45: **end if**

46: **CATE extraction (difference $\Rightarrow$ simplify):**
47: Compute $\hat{\mu}_0(\boldsymbol{x}), \hat{\mu}_1(\boldsymbol{x})$ by forward evaluation of the two heads
48: $\hat{\tau}(\boldsymbol{x}) \leftarrow \hat{\mu}_1(\boldsymbol{x}) - \hat{\mu}_0(\boldsymbol{x})$
49: $\textsc{SimplifyAlgebra}(\hat{\tau}(\boldsymbol{x}))$             $\triangleright$ cancel/factor common terms *after* the difference
50: **return** $(\hat{\mu}_0, \hat{\mu}_1, \hat{\tau}(\boldsymbol{x}), M)$

---

