# OpenReview forum: "CausalKANs: interpretable treatment effect estimation with Kolmogorov-Arnold networks"
_ICLR.cc/2026/Conference — Submitted to ICLR 2026_

### Official Review · Reviewer_aDc3 · 2025-10-25

**Soundness:** 3
**Presentation:** 3
**Contribution:** 2
**Rating:** 4
**Confidence:** 4

**Summary:**

This paper introduces causalKANs, a framework that transforms deep neural network estimators of conditional average treatment effects (CATEs) into Kolmogorov–Arnold Networks (KANs) to improve interpretability. By applying pruning and symbolic simplification, causalKANs produce closed-form, interpretable formulas while maintaining predictive accuracy and achieving a balance between performance and transparency.

**Strengths:**

- S1: Replacing MLPs with KANs in causal inference is well-suited, as KANs can model complex relationships with fewer parameters and improved interpretability, which supports more transparent estimation of treatment effects.

- S2: The proposed probability radar plots (PRPs) and partial dependence plots (PDPs) effectively illustrate the causal interpretability of the model in a clear and intuitive manner.

- S3: The paper is clearly written and easy to follow, with precise explanations and well-documented details throughout.

**Weaknesses:**

- W1: In Section 4, the paper’s main framework contribution includes a function simplification pipeline based on pruning and auto-symbolic search, which iteratively modifies model edges and performs revalidation on the validation set or even retraining. This procedure appears to impose substantial computational overhead, yet the paper provides no discussion of its complexity or efficiency. Including a complexity analysis similar to that used in tree-pruning methods would make the proposed approach appear more rigorous and practical.

- W2: The paper argues that simplifying KANs improves interpretability but implicitly introduces new assumptions about the dataset, such as the additivity assumption discussed at line 957. However, as shown in the IHDP-B experiment at line 395, replacing MLPs with KANs leads to poorer performance on heterogeneous datasets. The paper does not address how to validate these assumptions or guide model selection. Incorporating validation methods such as residual analysis or interaction tests could help assess their validity more systematically.

- W3: At line 1050, the paper uses test loss as the main model selection criterion. However, this metric may overemphasize outcome prediction accuracy and risk overfitting, rather than reflecting true treatment effect accuracy. As shown in Figure 11, PEHE appears to plateau as test loss increases, suggesting limited causal improvement despite better predictive fit.

**Questions:**

- Q1: At line 255, multiple regularization terms and $\lambda$ coefficients are introduced to control model complexity, but their selection is not discussed. Could the authors clarify how these values were determined or tuned?

- Q2: At line 1006, the experiment shows a weak correlation between model complexity and PEHE error. However, the definition of complexity lacks clear quantitative specification across different model settings. Could the authors provide more explanation, for instance, why “the weight penalty $\lambda$ contributes 1 if set to 0.01 and 2 otherwise”?

- Q3: At line 1407, replacing MLPs with KANs in the S-learner results in significantly slower performance compared to other models. Could the authors provide an explanation for this behavior?

- Q4: At line 1350 (Table 2), pruning appears to cause disastrous prediction errors on some datasets. Could the authors explain whether such degradation is expected or reasonable under their pruning criteria and stopping rules?

---

> ### Author Response · Authors · 2025-11-20
> **Response to reviewer aDc3**
>
> Thank you for your careful and constructive review. We appreciate the time dedicated in analyzing our work and the opportunity to clarify some points.
>
> >  W1. complexity analysis
>
> We agree that discussing computational efficiency strengthens the practical value. We will include a figure reporting the runtime that is in [the rebutal pdf (LINK)](https://anonymous.4open.science/r/causalkans_rebuttal-4EE0/rebuttal.pdf). These measurements, obtained on a CPU, show symbolic substitution can be comparable to or even exceed the KAN training time, especially for models with larger grids or higher complexity. We will provide a notion of the overhead introduced.
>
> > W2. new assumptions about the dataset [...] leads to poorer performance on heterogeneous datasets.
>
> We agree that it can introduce structural biases. This is why, throughout the paper, we treat KANs as a modeling option, not a universally superior alternative. Our evaluation reflects this: causalKANs are always compared directly with their neural counterparts, and the practitioner is expected to adopt the KAN version only when it does not incur a meaningful changes in MSE, or ITEs. For datasets like IHDP-B, where heterogeneity is high, the performance gap correctly signals that a KAN should not be selected despite its interpretability benefits. We will make this guidance more explicit.
>
> This philosophy aligns with standard practice in interpretable modeling, where structural constraints are used only when validation confirms they remain competitive with more flexible models [1].Other diagnostic tools are compatible with our approach, and we will discuss them as extensions. Our main contribution is the interpretability pipeline, which allow practitioners to set their own thresholds for accepting or rejecting a KAN substitution.
>
> > W3. overemphasize outcome prediction accuracy and risk overfitting
>
> We appreciate the reviewer’s observation. It is correct that outcome-prediction loss is not, in general, a perfect proxy for CATE accuracy, but all models evaluated in our work are potential-outcome regressors trained solely on factual losses. For this family of estimators, the standard practice is to select hyperparameters using validation prediction loss, since these methods do not expose orthogonalized or doubly robust objectives [2,3]. To ensure a fair comparison, causalKANs use the same training and selection rules.
>
> Regarding F.11, the plateau reflects intrinsic properties of outcome-regression methods, not an issue introduced by causalKANs. Still, we observe strong correlations (typically r > 0.8) between factual test loss and PEHE, indicating that loss remains a model-selection proxy for this class of learners. The slight flattening for the models is consistent with known behavior of non-orthogonalized estimators [2,4], where outcome prediction can improve faster than CATE error.
>
> We will add a discussion clarifying that causalKANs can use alternative selection criteria in such settings, but we did not apply them here to avoid asymmetric comparisons with the specific baselines studied.
>
> > Q1 how these values were determined or tuned?
>
> Those are hyperparameters that are tuned in the hp search. We prioritize higher values because they lead to simpler models.
>
> > Q2. Complexity lacks clear quantitative specification
>
> Our complecity score is a simple heuristic designed to summarize hyperparameter choices and also prioritize more interpretable models. In our search space, the spline regularization coefficient took only two values: 0.001 and 0.01. The latter produces simpler functions, we assigned it a lower complexity score. This metric is purely descriptive and it is included only to illustrate trends between model complexity and PEHE. We will add more discussion about this.
>
> > Q3.  S-learner results in significantly slower performance compared to other models
>
> S-KAN has a training time similar to other causalKANs, but S-learner is realy fast.
>
> > Q4. pruning causes disastrous prediction errors on some datasets
>
> We used the default pruning criterion because the goal is to evaluate pruning uniformly across all benchmark datasets; tuning it per-dataset would create an unfair comparison. Therefore , large errors can occur. A practitioner would either adjust the pruning threshold to limit changes in the estimated ITEs or simply reject pruning when it induces excessive error.
>
> Overall, we think we have addressed all raised concerns, provided detailed explanations for the experimental design choices, and clarified the behavior of the models and pruning strategies. We believe the clarifications resolve the concerns, and we kindly ask to consider updating the score accordingly.
>
> > Ref
>
> [1] Hastie 2017. Generalized additive models
> [2] Shalit, 2017. Estimating individual treatment effect: generalization bounds and algorithms. PMLR.
> [3] Shi, 2019. Adapting neural networks for the estimation of treatment effects.NeurIPS
> [4] Curth, 2021. Nonparametric estimation of HTE. PMLR.

---

> ### Comment · Area_Chair_Wbs6 · 2025-11-25
> **Please participate in discussions with authors and other reviewers asap**
>
> Please ensure you are actively participating in the discussion with authors.
>
> Additionally, I strongly encourage you to read the other reviews and discuss with your fellow reviewers. It is vital to compare perspectives and raise any remaining concerns now to give the authors a fair opportunity to respond.
>
> Based on these interactions, please update your reviews and finalize your decisions.
>
> Best, AC

---

### Official Review · Reviewer_M51h · 2025-10-29

**Soundness:** 3
**Presentation:** 3
**Contribution:** 2
**Rating:** 2
**Confidence:** 5

**Summary:**

The authors integrate Kolmogorov–Arnold Networks (KANs) into causal learning frameworks including the S-learner, T-learner, TARNet, and DragonNet to enhance interpretability while maintaining competitive accuracy in estimating conditional average treatment effects (CATEs). The proposed approach is empirically validated on benchmark datasets, including ACIC and IHDP.

**Strengths:**

This paper is well-written, and the interpretability analyses such as the radar plots and partial dependence plots are clear, intuitive, and effectively communicate the model’s behavior to the audience.

**Weaknesses:**

Despite its strengths, the paper exhibits several significant weaknesses.

Weakness 1 – Limited baseline comparisons: although the paper presents an interesting approach, it includes only a small set of baseline models—specifically TARNet, S-learner, T-learner, and DragonNet for evaluating estimation accuracy. However, numerous well-established alternatives exist, such as the DML, DR-learner, Ganite, and X-learner frameworks. For a paper without a solid theoretical contribution and aiming for a top-tier machine learning venue, it is essential to conduct a more comprehensive empirical comparison against a broader range of state-of-the-art methods to substantiate the claimed performance advantages.

Weakness 2 – Limited dataset diversity: The paper evaluates the proposed method solely on IHDP and ACIC, both of which are tabular, semi-synthetic datasets. This narrow experimental scope limits the generalizability and robustness of the findings. To strengthen the empirical contribution, the authors should consider incorporating a wider range of datasets, such as TCGA (additional tabular data), EEEC (text-based data), and Uganda (image data), following the precedent set by Wu et al. (2024), “Discret: Synthesizing Faithful Explanations for Treatment Effect Estimation,” Proceedings of Machine Learning Research, 235, 5359–97. Again, for a paper positioned as an applied contribution at a top-tier machine learning venue, it is crucial to demonstrate comprehensive experimentation across diverse data modalities.

Weakness 3- Limitations of KANs: following the weakness 2, one of the key motivations for using Kolmogorov–Arnold Networks (KANs) is their potential interpretability, as each edge represents a learnable univariate function that can be visualized and analyzed. However, the flexibility of per-edge spline functions may lead to overfitting or highly irregular learned mappings, particularly when applied to complex modalities such as image or text data. As discussed in Somvanshi et al. (2025), “A Survey on Kolmogorov–Arnold Networks,” ACM Computing Surveys, 58(2), 1–35, this challenge highlights the importance of validating the proposed framework on more diverse datasets including image and text-based benchmarks to ensure the robustness and scalability of causalKANs beyond simple tabular settings.

Weakness 4 – Limited novelty of CausalKANs: the core idea of enhancing interpretability in treatment effect estimation through additive neural architectures is not entirely novel. Similar concepts have been explored in prior work, such as Chen, K., Yin, Q., & Long, Q. (2023). “Covariate-Balancing-Aware Interpretable Deep Learning Models for Treatment Effect Estimation,” Statistics in Biosciences, 17(1), 132–150, where the authors employed neural additive models to estimate treatment effects. Since neural additive models (NAMs) can be viewed as a special case of Kolmogorov–Arnold Networks (KANs), the contribution of CausalKANs appears incremental rather than fundamentally new. A clearer discussion of how CausalKANs advance beyond existing NAM-based causal estimators would strengthen the paper’s methodological originality.

Weakness 5 – Limited interpretability across other estimator types: following the weakness 1, the main advantage of causalKANs seems to be improving interpretability for difference-in-mean estimators of treatment effects. However, it is not clear how this approach would enhance interpretability for other common estimators that rely on inverse probability weighting or doubly robust methods, such as the X-learner and DR-learner. Even for DragonNet with target regularization (see Section 3 of Shi, C., Blei, D., & Veitch, V. (2019), NeurIPS), the paper does not clearly explain how causalKANs make these models more interpretable. Providing a clearer discussion or examples showing how causalKANs could be applied to these estimators would make the paper stronger and more convincing.

Weakness 6 – Unclear reporting of evaluation metrics: the paper does not specify whether the reported empirical results reflect in-sample or out-of-sample errors. This distinction is important for assessing model generalizability and potential overfitting. It would strengthen the paper to report both in-sample and out-of-sample performance for all experiments, allowing readers to better understand the robustness of the proposed approach.

Overall, this is a clearly written and interesting paper. The proposed idea is promising, and the experiments are presented in an organized way. However, the paper could be significantly strengthened by either expanding the experimental evaluation to include a wider range of baseline methods and data modalities or by adding a solid theoretical component, such as a generalization error bound. In its current form, I do not think the paper meets the standard required for acceptance at ICLR 2026.

**Questions:**

Several points:

1.) Figure 2 presents box plots comparing MLP- and KAN-based models. From my perspective, there is no clear evidence that KANs consistently outperform MLPs, except perhaps for the T-learner. In some cases, such as DragonNet, KANs even appear to exhibit higher variance. It would be helpful if the authors could provide a deeper discussion of the observed bias–variance trade-off between MLP and KAN architectures in the context of treatment effect estimation.

2.) In Line 143, the paper states that $E[y \mid x, t]$ is an unbiased estimator of $\mu_t(x)$.  This appears to be a typo or conceptual inaccuracy, as an estimator should be data-driven  and cannot be expressed in terms of an expectation. The authors may wish to revise this statement  to clarify that $E[y \mid x, t]$ equals $\mu_t(x)$ under the standard causal assumptions,  rather than serving as its estimator.

---

> ### Author Response · Authors · 2025-11-20
> **Response to reviewer M51h (Part I)**
>
> We thank the reviewer for the careful reading of the manuscript and the constructive suggestions. Below we address each point in turn and will revise the paper accordingly.
>
> > W.1  it is essential to conduct a comparison against a broader range of state-of-the-art methods to substantiate the claimed performance advantages
>
> Our goal is not to show that causalKANs outperform a wide range of state-of-the-art methods, but that replacing neural backbones with KANs preserves performance while adding interpretability. We attempted to make this explicit (e.g., `line 369` “we intentionally restrict baselines to neural counterparts to test our central claim—comparable accuracy with improved interpretability”), but we will emphasize this more clearly in the final version.
>
> To address the reviewer’s point, we will add a brief discussion situating causalKANs relative to other CATE estimators explaining that benchmarking against the full spectrum of causal inference methods is orthogonal to our contribution and beyond the intended scope. Our focus is to demonstrate that causalKAN substitutions retain accuracy while enabling closed-form, interpretable estimators. We will clarify it in the introduction.
>
> >  W2. for a paper positioned as an applied contribution at a top-tier machine learning venue, it is crucial to demonstrate comprehensive experimentation across diverse data modalities
>
> We agree that additional dataset diversity can strengthen an empirical study. Our choice of IHDP A/B and the three ACIC regimes is intentional: these semi-synthetic benchmarks are the standard tabular datasets for CATE evaluation because they provide ground-truth potential outcomes, which is essential for quantitatively assessing PEHE and ATE error. Regarding data modalities,  we can provide some top-tier examples of the last 2 years in which CATE prediction is predominantly conducted on tabular, semi-synthetic datasets such as IHDP and ACIC [1–4 between others]. These benchmarks are designed specifically for causal inference and remain the most widely used setting for assessing CATE models. By contrast, evaluating treatment effects on modalities such as images or text is far less established and typically involves different problem formulations and assumptions. Extending our study to such modalities would therefore be outside the scope of this work—as it is for many recent CATE papers—and would not meaningfully inform our contribution, which focuses on interpretability for tabular CATE estimation.
>
> However, following the suggestion, and to expand the tabular evaluation, we added results on the NSLM dataset  (see table 5 [this anonymous repository](https://anonymous.4open.science/r/causalkans_rebuttal-4EE0/rebuttal.pdf) ) with similar resuts, and we are running TGCA and NEWS experiments, which show consistent trends. Nevertheless, extending the study to text or imaging modalities is outside the scope of this work: causalKANs are designed and evaluated as tabular CATE estimators, and incorporating unrelated modalities would not clarify the interpretability properties we aim to highlight. We will make this scope explicit in the introduction and frame multi-modal extensions as future work.
>
> > W3. spline functions may lead to overfitting or highly irregular learned mappings [...]  including benchmarks beyond simple tabular settings.
>
> Our work focuses on tabular CATE estimation, and extending causalKANs to images or text falls outside the intended scope of this paper. Within the tabular setting, we explicitly mitigate overfitting—an inherent risk in any flexible deep model—through several mechanisms: edge-activity and spline-coefficient regularization, early stopping, pruning guided by out-of-sample performance budgets, and optional symbolic substitution that further smooths the learned mappings. In practice, these steps produce sparser and smoother functions (in fig.18 and 19 of pdf in the anon. github we observe that causalKANs approximate better smooth functions than MLP), which aligns well with the structure of many real-world response surfaces, as also noted in prior KAN studies [5,6].
>
> Exploring other data modalities in KANs is still a line of work for many researchers, and such extensions for causal estimation is a promising direction for future research rather than a requirement for the present study.
>
>
> > References
>
> [1] Chen, et al. (2025). Covariate-balancing-aware interpretable deep learning models for treatment effect estimation.
>
> [2] Alaa,et al. (2023). Conformal meta-learners for predictive inference of individual treatment effects. NeurIPS
>
> [3]Wang, et al (2023). Optimal transport for treatment effect estimation. NeurIPS
>
>  [4] Guo, et al(2023). Estimating heterogeneous treatment effects: Mutual information bounds and learning algorithms.  PMLR.
>
> [5] Wang et al (2025). Kolmogorov–Arnold-Informed neural network: [...]. Computer Methods in Applied Mechanics and Engineering
>
> [6]Liu et al (2024). Kan: Kolmogorov-arnold networks.

---

> > ### Author Response · Authors · 2025-11-20
> > **Response to reviewer M51h (Part II)**
> >
> > > W4. clearer discussion of how CausalKANs advance beyond existing NAM-based causal estimators would strengthen the paper’s methodological originality
> >
> > In the revision, we will expand the related-work section to position causalKANs more explicitly relative to NAMs and other GAM-like approaches. While NAMs offer decomposable architectures, they do not provide closed-form symbolic expressions nor a principled pipeline for transforming neural estimators into interpretable representations. The contribution of causalKANs is precisely this pipeline: KAN-ification of causalNNs, pruning guided by performance budgets, and symbolic substitution that yields analytic CATE formulas, not just interpretable structures.
> >
> > Moreover, our pipeline regularizes the functional behavior learned by neural networks, which can exhibit spiking or irregular patterns (see Fig. 19 in the anonymous repository). The simplification steps produce smoother, more stable mappings, which directly enhances interpretability and faithfulness. We will make these distinctions explicit to better articulate how causalKANs advance beyond existing NAM-style causal models.
> >
> > >W5. how this approach would enhance interpretability for other common estimators? clearer discussion or examples showing how causalKANs could be applied
> >
> > We agree that a clearer discussion of how causalKANs extend interpretability to other CATE estimators would strengthen the paper. We will add an appendix section explaining, for common methods such as DR-learners, R-learners, and other meta-learners, which components can be replaced by KANs and what type of interpretability this enables.
> >
> > A key point is that interpretability is task-specific: different estimators learn different functional components, and the practitioner must decide which function they want to interpret. For instance, in IPW the only learnable part is the propensity model; replacing it with a KAN allows full interpretability of the propensity function in the logit space, but the ATE computation itself remains unchanged. By contrast, the X-learner contains two regression blocks, both of which can be “KAN-ified.” The potential-outcome regressions become fully auditable, and, more importantly, the final CATE regressor can be directly inspected, analogous to T-KAAM, yielding an interpretable effect curve as a function of covariates.
> >
> > In summary, causalKANs can enhance interpretability across many estimators, but the kind of interpretability depends on which sub-model the estimator learns and what aspects of the causal mechanism the practitioner aims to understand. We will make this explicit in the revised manuscript and provide concrete examples.
> >
> > W6. Unclear reporting of evaluation metrics: the paper does not specify whether the reported empirical results reflect in-sample or out-of-sample errors
> >
> > We thank the reviewer for noting this ambiguity. Although PEHE is formally defined over the test set in the manuscript (line 349) , we agree that the tables and figures should explicitly state that all reported metrics—PEHE and ATE error—are computed strictly out-of-sample on held-out test points unseen during training and validation. In the revision we will standardize captions and table descriptions to make this explicit. Concretely: (i) hyperparameter tuning and early stopping use a separate validation split; (ii) only test-set metrics are reported in the main text; and (iii) no in-sample errors appear in the current version.
> >
> > To improve transparency, we will additionally report training-set metrics in the appendix, which will illustrate how pruning and symbolic substitution affect train–test gaps and help readers assess potential overfitting.
> >
> > > Q1. there is no clear evidence that KANs consistently outperform MLPs
> >
> > Our intended message (which we will clarify) is that across architectures and datasets, at least one causalKAN variant is either the best model or statistically indistinguishable from the best neural model in PEHE/ATE, while admitting analytic inspection, not to achieve the best performance.
> >
> > > Q2.  an estimator should be data-driven and cannot be expressed in terms of an expectation
> >
> > We thank the reviewer for catching this. The current phrasing is indeed inaccurate: an expectation is a population quantity, not an estimator. Our intention was to state that, under the standard causal identification assumptions, $\mu_t(x) = E[y|t,x]$
> > and therefore any unbiased estimator of this conditional expectation is also an unbiased estimator of $\mu_t(x)$. We will revise the paper and remove the misleading phrasing that presents the expectation itself as an estimator.
> >
> > >
> > Thank you once again for the constructive and detailed feedback. We believe the revisions outlined above address all concerns and substantially strengthen the clarity, scope, and transparency of the work. We would be grateful if, after reviewing these updates, the reviewer find the manuscript improved and consider raising the score accordingly.

---

> ### Comment · Area_Chair_Wbs6 · 2025-11-25
> **Please participate in discussions with authors and other reviewers asap**
>
> Please ensure you are actively participating in the discussion with authors.
>
> Additionally, I strongly encourage you to read the other reviews and discuss with your fellow reviewers. It is vital to compare perspectives and raise any remaining concerns now to give the authors a fair opportunity to respond.
>
> Based on these interactions, please update your reviews and finalize your decisions.
>
> Best, AC

---

### Official Review · Reviewer_CfPa · 2025-11-01

**Soundness:** 3
**Presentation:** 3
**Contribution:** 2
**Rating:** 4
**Confidence:** 3

**Summary:**

The paper presents a novel framework for causal inference, transforming opaque neural CATE estimators into closed-form formulas using Kolmogorov-Arnold Networks (KANs). The claim is extracting human-readable (accept/reject gates, explicit budgets) causal effect formulas from neural networks. The algorithm works with standard architectures e.g. s-learner, r-learner, DragonNet. The paper claims competitive predictive performance that matches or exceeds neural baselines. The authors also provide analysis and visualization across multiple benchmark datasets.

**Strengths:**

- The paper proposes an innovative algorithm with KANs which replace neural networks in causal learning architectures. This makes the algorithm more interpretable
- The algorithm with KAN generally performs better or matches the performance of neural networks, tested on multiple datasets
- The overall approach has significant potential, and i'd be interested to know how KAN can contribute to solving novel tasks that are difficult for NN based causal learning

**Weaknesses:**

- The paper may be seen as slightly incremental due to the plug-in of KANs to replace neural networks, and the algorithm itself didn't have other innovations. The contribution seems more towards interpretation and validation in experiments
- More datasets quoted in previous work may help strengthen the paper

e.g.
[1] Rankability-enhanced revenue uplift modeling framework for online marketing. He, Bowei and Weng, Yunpeng and Tang, Xing and Cui, Ziqiang and Sun, Zexu and Chen, Liang and He, Xiuqiang and Ma, Chen.

- ablation studies and more examples of interpretability offered by KAN's specific to the task/dataset will strengthen the paper.

**Questions:**

- could more dataset comparison be included in the paper
- I think the interpretation with accept/reject gate etc. is a bit less well explained. Could you give examples and interpret the architecture or signals given a dataset and task?
- are there tasks you think KAN with causal learning/HTE may particularly excel at, ones that are challenging for NNs?

---

> ### Author Response · Authors · 2025-11-20
> **Response to reviewer CfPa (part I)**
>
> We thank the reviewer for the helpful comments. Below we address the main concerns and indicate how the paper will be improved.
>
> > W1. The paper may be seen as slightly incremental due to the plug-in of KANs to replace neural networks, and the algorithm itself didn't have other innovations. The contribution seems more towards interpretation and validation in experiments.
>
> We acknowledge that our contribution is primarily practical rather than theoretical, and we are explicit about this in the manuscript. However, we believe the work addresses an important and currently unmet need in causal inference. Our goal is to systematically introduce Kolmogorov–Arnold Networks into the CATE estimation ecosystem and to demonstrate how replacing standard neural backbones with KANs fundamentally changes what practitioners can do: namely, obtain closed-form, auditable effect models while preserving the strong empirical performance of widely used causal neural networks.
>
> The novelty therefore does not lie in proposing a new causal architecture, but in adapting KANs—originally developed outside causal inference—to a setting where interpretability is essential. We do this through a model-agnostic pipeline that replaces the neural backbones of standard CATE estimators (meta-learners, TARNet, DragonNet) with KANs and augments them with pruning, symbolic substitution, and visualization tools tailored specifically for potential outcomes and heterogeneous treatment effects. This adaptation substantially improves the transparency and auditability of deep CATE models, which is particularly relevant in high-stakes domains such as healthcare, economics, and public policy.
>
> From this perspective, the contribution is not the architectural substitution alone but the demonstration that KANs can be effectively operationalized for causal inference, enabling interpretable CATE estimation without sacrificing accuracy. Overall, the work provides a coherent foundation for interpretable causal deep learning, one that aims to support future research and responsible adoption rather than isolated architectural novelty.
>
> > W2. More datasets quoted in previous work may help strengthen the paper.
>
> We agree that dataset diversity. Our experimental design follows the conventions in the causal inference literature, where IHDP and ACIC constitute the standard semi-synthetic benchmarks for evaluating CATE models. These are not two datasets but five distinct benchmark settings (IHDP A/B and three ACIC regimes), and most recent works reporting state-of-the-art results evaluate on these settings [1–4]. We agree that additional datasets further strengthen the practical contribution. Motivated by this, we extended our evaluation to the NSLM dataset [5], for** which results are included in Table 5 of the PDF** in [this anonymous repository](https://anonymous.4open.science/r/causalkans_rebuttal-4EE0/rebuttal.pdf). The conclusions remain consistent: causalKANs achieve performance comparable to or better than their neural counterparts while providing interpretable closed-form CATE expressions. Although time constraints prevented full integration before the deadline, we are currently running experiments on the NEWS dataset introduced in [6] as well as the TGCA dataset [7], and preliminary results indicate similar behavior.
>
> Overall, we believe the current evaluation covers the main benchmarks used in prior work while the added NSLM results and ongoing experiments further demonstrate that causalKANs generalize well beyond the classical IHDP/ACIC.
>
> > W3. ablation studies and more examples of interpretability offered by KAN's specific to the task/dataset will strengthen the paper.
>
> We have expanded both ablations and interpretability analyses. First, Section C.1 already serves as an ablation study: we vary depth, spline grids, regularization, and model complexity, and report how these choices correlate with PEHE across datasets. These results show that simpler KAN architectures often perform as well as more complex ones, supporting their use for interpretability. Second, we quantify how each step of the pipeline—KAN training, pruning, symbolic substitution—affects all metrics (MSE, ATE error, PEHE). The file pipeline_metrics.pdf in the [anonymous repository](https://anonymous.4open.science/status/causalkans_rebuttal-4EE0) provides these step-by-step results for all benchmarks.
> Finally, our SCM experiments offer direct interpretability validation: with known structural equations, causalKANs recover the true functional shapes and closely match the ground-truth CATE. We can observe those experiments in the rebuttal.pdf file of [this anonymous repository](https://anonymous.4open.science/r/causalkans_rebuttal-4EE0/rebuttal.pdf). If the reviewer has a specific ablation in mind, we would be interested in adding it.

---

> > ### Author Response · Authors · 2025-11-20
> > **Response to CfPa (Part II)**
> >
> > (we continue here)
> >
> > > I think the interpretation with accept/reject gate etc. is a bit less well explained. Could you give examples and interpret the architecture or signals given a dataset and task?
> >
> > **We will expand the explanation of the accept/reject mechanism**. Appendix D already provides the full pipeline, but we will clarify how it operates in practice. At each simplification step (pruning or symbolic substitution), we recompute the factual loss on a validation set and compare it with the loss of both the unsimplified KAN and the original neural model. If the degradation exceeds a user-specified budget, the change is rejected. This ensures that interpretability steps never produce a surrogate model that diverges too far from the trained predictor.
> >
> > The behavior of the gate depends on task-specific parameters such as the pruning threshold and the minimum R2
> >  for symbolic substitution. In our benchmarks we use reasonable defaults, but in real applications practitioners can tune these thresholds according to the acceptable accuracy–interpretability trade-off for their domain. In the revised version we will add a concrete illustration of how these gates behave on a dataset, and how they determine whether a simplification is accepted or not.
> >
> > > are there tasks you think KAN with causal learning/HTE may particularly excel at, ones that are challenging for NNs?
> >
> > In the paper we note that, although KANs share universal approximation guarantees with standard neural networks, their practical strengths emerge when the underlying data-generating process aligns with their inductive biases. KANs are most effective when the response surfaces are smooth and approximately decomposable into low-dimensional additive components. This is consistent with observations from physics-informed and PDE benchmarks, where KANs outperform MLPs under smooth, structured dynamics [8], and with the original KAN work, which highlights the prevalence of smooth and sparse functional behavior in real-world systems [9]. From a causal perspective, several works argue that treatment-effect functions can often be simpler than the potential outcome surfaces they derive from [10]. In such cases, causalKANs can exploit this structure: pruning and symbolic substitution naturally act as regularizers that favor simple, interpretable functional forms, whereas MLPs may not bias toward these representations. This behavior is evident in our SCM experiment, where the true structural equations are smooth and additive; symbolic KANs recover the correct functional shape and outperform both MLPs and non-symbolic KANs. In practice, assumptions such as smoothness, separability, or approximate additivity are only partially testable, so we view KANs as particularly promising for domains where these properties are plausible—such as physics-based modeling, biomedical dose–response surfaces, and policy settings with structured covariate effects. Systematic comparisons between KAN and MLP backbones, with and without pruning, remain a practical way to probe which inductive bias aligns better with a given task. **we will add this discussion to the paper**
> >
> > In summary, we (i) clarified the practical contribution beyond incrementality, (ii) added NSLM results and ongoing NEWS/TGCA evaluations, (iii) expanded ablations and interpretability experiments (including SCM identifiability), and (iv) improved the explanation of the accept/reject mechanism. We believe these revisions fully address your concerns and respectfully ask the reviewer to consider raising the score if their main concerns have been addressed.
> >
> > > References
> >
> > [1] Chen, et al. (2025). Covariate-balancing-aware interpretable deep learning models for treatment effect estimation.
> >
> > [2] Alaa,et al. (2023). Conformal meta-learners for predictive inference of individual treatment effects. NeurIPS
> >
> > [3]Wang, et al (2023). Optimal transport for treatment effect estimation. NeurIPS
> >
> >  [4] Guo, et al(2023). Estimating heterogeneous treatment effects: Mutual information bounds and learning algorithms.  PMLR.
> >
> > [5] Carvalho et al. (2019). Assessing treatment effect variation in observational studies: Results from a data challenge. Observational Studies,
> >
> > [6]Johansson,et al. (2016, June). Learning representations for counterfactual inference.PMLR.
> >
> > [7] Schwab, P., Linhardt, L., Bauer, S., Buhmann, J. M., & Karlen, W. . Learning counterfactual representations for estimating individual dose-response curves. AAAI.
> >
> > [8] Wang et al (2025). Kolmogorov–Arnold-Informed neural network: A physics-informed deep learning framework for solving forward and inverse problems based on Kolmogorov–Arnold Networks. Computer Methods in Applied Mechanics and Engineering
> >
> > [9]Liu et al (2024). Kan: Kolmogorov-arnold networks.
> >
> > [10] Curth & van der Schaar (2021). On inductive biases for heterogeneous treatment effect estimation. NeurIPS

---

> ### Comment · Area_Chair_Wbs6 · 2025-11-25
> **Please participate in discussions with authors and other reviewers asap**
>
> Please ensure you are actively participating in the discussion with authors.
>
> Additionally, I strongly encourage you to read the other reviews and discuss with your fellow reviewers. It is vital to compare perspectives and raise any remaining concerns now to give the authors a fair opportunity to respond.
>
> Based on these interactions, please update your reviews and finalize your decisions.
>
> Best, AC

---

### Official Review · Reviewer_Gp2j · 2025-11-01

**Soundness:** 4
**Presentation:** 4
**Contribution:** 3
**Rating:** 8
**Confidence:** 4

**Summary:**

The paper presents CausalKAN framework with code contributions that address treatment effect estimation with interpretable results. This has applications in sensitive decision making systems to understand covariates relevant to individual treatment effects.

**Strengths:**

Originality: While the paper is well written and the interpretability angle is valuable, the core technical ideas appear close to KANITE [1], which targets performance rather than interpretability.

Quality & clarity: With its key contribution as interpretable results, I found the exposition clear and well organized. The manuscript delineates the symbolic simplification and pruning techniques in a reproducible way (definitions, criteria, and algorithmic steps are explicit) and demonstrates how they lead to interpretable models via semi-synthetic experiments.

Significance: The work is a useful step toward interpretable effect estimation for sensitive decision-making (e.g., healthcare, lending, education). While the technical novelty is not groundbreaking, the paper’s pipeline makes ITE models more transparent and could aid auditing and deployment. The current evidence (largely semi-synthetic) is promising but would benefit from stronger identifiability statement for interpretable symbolic regression (see below for details).

[1] E. Mehendale, A. Thorat, R. Kolla, and N. Pedanekar. Kanite: Kolmogorov-arnold networks for ite estimation. arXiv preprint arXiv:2503.13912, 2025

**Weaknesses:**

*Evaluation for interpretability/faithfulness.* I recommend including symbolic CATE benchmarks: construct SCMs with known closed-form symbolic CATE $\tau(x)$ and report whether KAN’s simplified/pruned representation recovers the underlying expression (up to algebraic equivalence) alongside prediction error. This directly evaluates faithfulness, which is key for trustworthy use in sensitive decisions, and would make the interpretability claim much more convincing.

**Questions:**

See above. I would be interested to see how CausalKAN address the faithfulness of the interpretable results.

---

> ### Author Response · Authors · 2025-11-20
> **Response to reviewer Gp2j**
>
> We thank the reviewer for the careful reading, the positive assessment and the constructive suggestions.
> We would like to comment how we will address each of these suggestions in the revised version of the paper.
>
> >  the core technical ideas appear close to KANITE, which targets performance rather than interpretability.
>
> We agree that our architectural contribution is incremental and that some components overlap with KANITE. We have been transparent about this in the manuscript and already note that, to the best of our knowledge, KANITE is the first work using KANs for causal estimation. **We will make this acknowledgement more prominent in the Introduction and Related Work**. However, our contribution has a different purpose: KANITE is designed to improve predictive performance for multi-arm ITE estimation, whereas causalKANs are designed to make established CATE estimators interpretable, auditable, and suitable for deployment. In many real-world settings—healthcare, public policy, economics—CATE models support consequential decisions, and in these domains **transparency is not secondary to accuracy but a requirement for responsible use**. Our framework addresses this need directly by (i) providing a model-agnostic pipeline to “KAN-ify’’ canonical CATE architectures (meta-learners, TARNet, DragonNet), and (ii) introducing pruning and symbolic substitution with explicit performance budgets, yielding closed-form CATE expressions and interpretable visualizations. The goal is therefore not to create a new causal architecture, but to enable widely adopted causal neural models to be used in settings where interpretability and trustworthiness are as important as predictive performance
>
> > I recommend including symbolic CATE benchmarks: construct SCMs with known closed-form symbolic CATE and report whether KAN’s simplified/pruned representation recovers the underlying expression (up to algebraic equivalence) alongside prediction error.
>
> We appreciate the reviewer’s insightful suggestion to evaluate faithfulness directly. In the revised submission, we will add a focused interpretability experiment based on simple SCMs with known structural equations. Because the true functions are available, we can compare them against the estimates produced by S-KAAM and T-KAAM. In both SCMs, the ground-truth response functions lie within the additive function class recoverable by KAAMs, and we observe that the learned curves closely match the analytic forms: polynomial terms are correctly identified, and the shape of the recovered functions aligns with the true mechanisms. As expected, identification holds up to an additive constant for each potential outcome, but this does not affect CATE, which is fully recovered. This can be seen in Figures 18B–C and 19D of the supplementary PDF in [this anonymous repository](https://anonymous.4open.science/r/causalkans_rebuttal-4EE0/rebuttal.pdf), where the symbolic CATE matches the ground truth almost exactly. Notably, in these experiments pruning and symbolic substitution do not degrade performance; they improve it. The simplified KANs produce smoother functions and yield smaller errors than the unsimplified models. This reflects the inductive bias toward smoothness that KANs introduce, as discussed in [1], and which can be advantageous when the underlying data-generating process is itself smooth—a property that holds in many real-world domains.
>
> In summary, we provide: (i) a direct faithfulness check using SCMs with known analytic forms, (ii) visual and quantitative evidence that symbolic causalKANs recover the true causal mechanisms in identifiable settings, and (iii) a demonstration that the interpretability steps do not degrade fidelity and can even improve it. We hope this experiment aligns with the reviewer’s concerns and helps illustrate the faithfulness properties of causalKANs.
>
> We thank the reviewer once again for the constructive feedback, which has helped us strengthen both the clarity and the empirical depth of the work, and we hope that the added material satisfactorily addresses the questions raised.
>
> > References
>
> [1] Liu, Z., Wang, Y., Vaidya, S., Ruehle, F., Halverson, J., Soljačić, M., ... & Tegmark, M. (2024). Kan: Kolmogorov-arnold networks. arXiv preprint arXiv:2404.19756.

---

> ### Comment · Area_Chair_Wbs6 · 2025-11-25
> **Please participate in discussions with authors and other reviewers asap**
>
> Please ensure you are actively participating in the discussion with authors.
>
> Additionally, I strongly encourage you to read the other reviews and discuss with your fellow reviewers. It is vital to compare perspectives and raise any remaining concerns now to give the authors a fair opportunity to respond.
>
> Based on these interactions, please update your reviews and finalize your decisions.
>
> Best, AC

---

### Author Response · Authors · 2025-12-02
**Final Statement to the Area Chair with Reviewer feedback (part I)**

We would like to make some final comments about the reviewing process, highlighting that we think we have addressed all the concerns raised by the reviewers that we believe are sensible to address. On the other hand, there exist other weaknesses that we think are not within the scope of our paper, and we will justify this thought. In general, we would like to stress that **all the important concerns of the reviewers were suitable to be resolved in this revision period**, and that we have a justification for not addressing one of the major concerns of the most pessimistic reviewer (M51h): _adding multimodality_, which we think is out of the scope of this paper. But let us structure our thoughts.

First of all, we thank the reviewers for their positive feedback. All of them have highlighted common strengths in our paper

- Significance in sensitive decision-making domains: _The work is a useful step toward interpretable effect estimation for sensitive decision-making_ (reviewer Gp2j); _The overall approach has significant potential_... (reviewer CfPa); _...improved interpretability, which supports more transparent estimation of treatment effects._ (reviewer aDc3)

- Clarity in both writing and interpretability results: _With its key contribution as interpretable results, I found the exposition clear and well organized [...] demonstrates how they lead to interpretable models via semi-synthetic experiments_ (reviewer Gp2j); _This paper is well-written, and the interpretability analyses such as the radar plots and partial dependence plots are clear, intuitive, and effectively communicate the model’s behavior to the audience_ (reviewer M51h); _effectively illustrate the causal interpretability of the model in a clear and intuitive manner [...] is clearly written and easy to follow, with precise explanations and well-documented details throughout_ (reviewer aDc3)

Secondly, we really think that the **current version** has been strengthened based on the insightful suggestions for improvement given by the reviewers. We include here a **list of important changes**, denoted in orange (replacement) and blue (addition) that can be consulted in the curren PDF.

- **More datasets** NSLM [1], NEWS [2], and TCGA [3] datasets have been included in the study, which can be observed in the modified **Table 1**, Table 2 and Table 3. These datasets showed consistent results with the first version, and it can be observed that causalKANs achieved comparable or better performance in 7 of the 8 datasets. This concern is shared by reviewers CfPa, M51h and aDc3, so we think it has strengthened the paper.

- **More practical discussion.** Required discussion about **faithfulness** (raised by Gp2j) added in section 5.3 and Appendix C.4; about pipeline computational overhead (aDc3) in Appendix C.5; about when the inductive bias of using KANs should outperform MLP (raised by CfPa), added in lines 317-328, and also supported by the experiment of Section 5.3; deeper relationships with KANITE and causal NAMs in lines 121-136; we clearly report **out-of-sample metrics** (raised by M51h) in Table 1; about future work in other modalities (raised by M51h) in lines 530-535; about extending causalKANs to other modalities (raised by M51h) in Appendix B.1; about complexity and intuitions of ablation studies (raised by aDc3) in Appendix C.2; and more details about pipeline metrics and how accept/reject gates should be applied (raised by CfPa) have been included in Appendix D and Appendix C.3.3.

- **Clarification of the scope** we highlighted that our objective is to work with **tabular data** (regarding M51h's concern), and that our benchmarking experiment **does not aim to show that causalKANs outperform causalNNs**, but only to show **competitive performance**, in order to demonstrate that approximation capabilities of ITE and ATE hold (raised by M51h).
- **Faithfulness experiments**. As suggested by Gp2j, we added synthetic experiments to show how well causalKANs capture the true generating equation, showing that causalKANs can improve both performance and interpretability when the true functions lie in the class that causalKANs can model (see Fig 6 and discussion of Section 5.3)

---

> ### Author Response · Authors · 2025-12-02
> **Final Statement to the Area Chair with Reviewer feedback (part II)**
>
> Lastly, we want to discuss the **only concerns we have not addressed**, which we think are the main reasons **why reviewer M51h decided to give a negative score**: the addition of datasets of other modalities beyond tabular data. The reviewer suggested that, _for a top-tier machine learning venue_, our method should show the best performance compared with other methods _such as the DML, DR-learner, Ganite_, and that _it is crucial to demonstrate comprehensive experimentation across diverse data modalities_. While we agree that adding more datasets could improve the practical strength of the paper, and we included more datasets (see above), we kindly disagree with the reviewer that we should include more modalities for a top-tier conference, since we can cite several works in causal inference that **only use tabular data** [4,5,6,7], and have been published in these venues in the **last 2 years**. In addition, the task of performing causal inference on other modalities is still an open problem in the community.
>
> On the other hand, we also explained that **comparing our framework with other estimators** is **out of the scope** of our paper because our aim is to demonstrate that we propose a pipeline in which substituting MLP by KANs does not hurt causal performance, and we can obtain interpretable results. Although the pipeline supports other estimators (Appendix B.1), it is not the objective to show all of them, but to give some examples for practitioners to know how to proceed and use our pipeline. Also, our pipeline supports the decision of rejecting the causalKAN because the performance is bad, since we consider the use of KANs only as a modeling option (see Appendix D).
>
> For all these reasons, we strongly think that **we have addressed the main reasons why some of the reviewers gave a negative score to the paper**, and that they could have raised their scores had they had the opportunity. We hope the AC considers all the improvements that we have made to our paper, and we thank all the reviewers for helping us construct a more solid paper with **strengthened contributions**, experimental insights and practical application.
>
> [1] Carvalho et al. (2019). Assessing treatment effect variation in observational studies: Results from a data challenge. Observational Studies,
>
> [2] Johansson et al. (2016, June). Learning representations for counterfactual inference. PMLR.
>
> [3] Schwab, P., Linhardt, L., Bauer, S., Buhmann, J. M., and Karlen, W. Learning counterfactual representations for estimating individual dose-response curves. AAAI.
>
> [4] Alaa et al. (2023). Conformal meta-learners for predictive inference of individual treatment effects. NeurIPS
>
> [5] Wang et al. (2023). Optimal transport for treatment effect estimation. NeurIPS
>
> [6] Guo et al. (2023). Estimating heterogeneous treatment effects: Mutual information bounds and learning algorithms. PMLR.
>
> [7] Javaloy, A., Sánchez-Martín, P., and Valera, I. (2023). Causal normalizing flows: from theory to practice. Advances in Neural Information Processing Systems, 36, 58833-58864.

---

### Meta-Review · Area_Chair_ikND · 2026-01-04

**Summary:**

This paper applies the KAN structure into the task of causal effect estimation. Aside from achieving the estimation accuracy, causalKAN is capable of achieving the interpretation in some extent.

However, after reading the (revised) manuscript, my concerns fall in some aspects:
- [-] Lack of enough motivation on the trade-off between interpretation and estimation performance. For instance, as tree-based methods (CausalTree or CausalForest) can offer enough interpretation without degrading the performance, why the causalKAN should be developed?  Meanwhile, how to define and quantify the interpretation of a causal effect estimation model is also missing in the starting sections of the paper. Therefore, this paper is not motivated well. Although introducing interpretation into causal effect models is interesting, either the choice of the KAN backbone and the detailed comparison with tree-based methods is missing.

Therefore, given the lack of justification on the issue I have mentioned above, I recommend to reject.

**Reviewer Concerns:**

Reviewer M51h's concerns are not addressed at all.

**Reviewer Scores:**

No reviewer will change their score given the concurrent rebuttal.

---

### Decision · Program_Chairs · 2026-01-26

Reject